# SEQUOIA: Scalable and Robust Speculative Decoding

**Zhuoming Chen**[1*]   **Avner May**[2*]   **Ruslan Svirschevski**[3,4*]
**Yuhsun Huang**[1]   **Max Ryabinin**[2]   **Zhihao Jia**[1]   **Beidi Chen**[1,5]
[1]Carnegie Mellon University   [2]Together AI   [3]Yandex
[4]National Research University Higher School of Economics   [5]FAIR, Meta
zhuominc@andrew.cmu.edu, avner@together.ai, ruslansv@gmail.com
yuhsunh@andrew.cmu.edu, mryab@together.ai, {zhihaoj2,beidic}@andrew.cmu.edu

## Abstract

As the usage of large language models (LLMs) grows, it becomes increasingly important to serve them quickly and efficiently. While speculative decoding has recently emerged as a promising direction for accelerating LLM serving, existing methods are limited in their ability to scale to larger speculation budgets and adapt to different hyperparameters. This paper introduces SEQUOIA, a scalable and robust algorithm for speculative decoding. To improve scalability, SEQUOIA introduces a dynamic programming algorithm to find an optimal tree structure for the speculated tokens. To achieve robust speculative decoding, SEQUOIA uses a novel sampling and verification method that outperforms prior work across different decoding temperatures. SEQUOIA improves the decoding speed of Llama2-7B, Llama2-13B, and Vicuna-33B on an A100 GPU by up to $4.04\times$, $3.73\times$, and $2.27\times$. To serve Llama3-70B-Instruct on a single L40 GPU through offloading, SEQUOIA reduces the per-token decoding latency to 0.60 s/token, $9.5\times$ faster than DeepSpeed-Zero-Inference. The code is available at `https://github.com/Infini-AI-Lab/Sequoia`.

## 1   Introduction

As large language models (LLMs) gain widespread adoption [3, 43, 7], efficiently serving these LLMs becomes increasingly important. However, accelerating LLM inference is challenging since generating a single new token requires accessing all parameters of the LLM [34]. As a result of this I/O bottleneck, the hardware is poorly utilized during generation. This problem is exacerbated in both small-batch and offloading-based inference settings, where generating one token takes as much time as processing a prompt with hundreds or thousands of tokens on modern GPUs.

To address this challenge, recent work has introduced *speculative decoding* to accelerate LLM inference while preserving the LLM's output distribution [24, 5, 28, 40]. These approaches leverage one or multiple *draft models* to predict the LLM's output; the predictions are organized in a *token tree*, whose nodes represent different sequences of speculated tokens. The correctness of these speculated tokens is then *verified in parallel* through a single forward pass of the LLM. Using a token tree—instead of a sequence—can increase the number of tokens accepted by the LLM by providing several options for each token position.

While there are substantial studies on tree-based speculative decoding methods [28, 40], we see in our experiments that they have a couple of limitations. First, we observe that existing token tree construction algorithms perform well for small token trees but are sub-optimal for large tree sizes. For example, SpecInfer constructs a token tree using $k$ independent sequences, a topology that is bounded by the expected number of tokens it can accept, regardless of the tree size (Figure 1). Second, we observe that existing token tree sampling and verification algorithms are unable to perform well across inference hyperparam-

---

[*]Equal contribution

38th Conference on Neural Information Processing Systems (NeurIPS 2024).

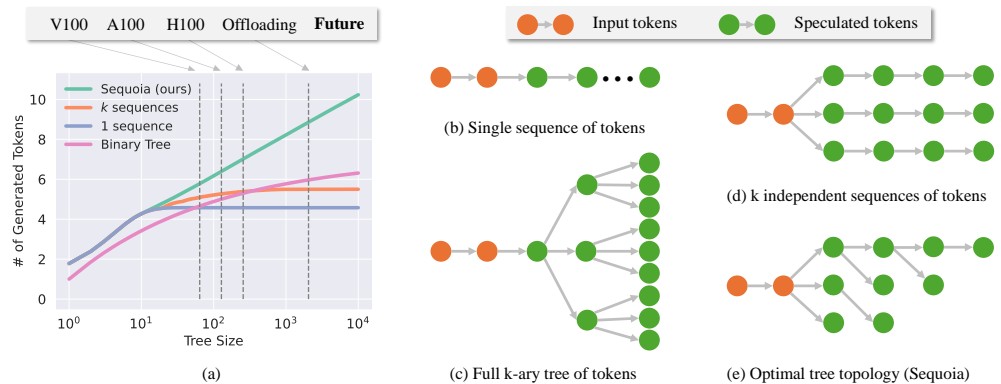

Figure 1: SEQUOIA is a scalable method for speculative decoding. Left: SEQUOIA tree construction algorithm is able to generate trees whose average number of generated tokens (after verification) continues to grow with the tree size while existing tree structures asymptote. This allows SEQUOIA to perform much better than existing methods in very memory-bound regimes like offloading. Right: A visualization to contrast SEQUOIA tree structure with other common handcrafted ones.

eter configurations; for example, SpecInfer [28] and SpecTr [40] often perform poorly at low temperatures (Figure 3) since they can repeatedly sample an incorrect token with high draft model probability.

In this paper, we aim to answer the following research question: *how can we design an optimal tree-based speculative decoding method to maximize speedups on modern hardware?* Realizing this goal requires addressing several technical challenges. First, for any tree size and depth, we must be able to efficiently search the exponentially large space of tree topologies to find the one that maximizes the expected number of generated tokens. Second, we must design a tree sampling and verification procedure that performs well across inference hyperparameters, avoids repeatedly sampling incorrect tokens, and maintains the correct output distribution.

This paper introduces SEQUOIA, a scalable and robust speculative decoding algorithm. As shown in Figure 1, SEQUOIA can attain up to $9.5\times$ speedups over incremental decoding and introduces several key techniques to address the aforementioned challenges.

- In Section 3.1, to solve the first challenge, we formulate tree construction as a constrained optimization problem and employ a dynamic programming algorithm to discover the *optimal* speculative token tree. Theoretically and empirically, we demonstrate that the number of tokens generated with this tree structure is unbounded, growing roughly logarithmically with the tree's size.
- In Section 3.2, to address the second challenge, we build upon the SpecInfer [28] algorithm by performing sampling *without replacement* from the draft model—thereby preventing the draft model from making the same mistake twice, while maintaining the target model's output distribution. We prove that this new sampling and verification method can attain high acceptance rates at both high and low temperatures and validate this claim empirically.

In Section 4, we perform extensive end-to-end experiments and ablation studies to demonstrate the effectiveness of SEQUOIA. We implement SEQUOIA on top of Hugging Face [45] with CUDA Graphs [31, 32]. We show that SEQUOIA achieves up to $4.04\times$ speedup for Llama2-7B on a single A100 GPU and $9.5\times$ for Llama3-70B-Instruct in the offloading setting on an L40 GPU. The latency of Llama3-70B-Instruct offloading on L40 can be reduced to 0.60 s/token with SEQUOIA while the inference speed of state-of-the-art offloading system (DeepSpeed-Zero-Inference [2]) is 5.7 s/token. We also present ablation studies to show that: (1) the SEQUOIA tree structure can generate up to $33\%$ more tokens per decoding step compared to $k$ independent sequences (tree size $\leq 512$), demonstrating better scalability; (2) the SEQUOIA sampling and verification algorithm is robust to the choice of hyperparameters (temperature, top-$p$), providing up to $65\%$ and $27\%$ speedup compared to SpecInfer and top-$k$ sampling and verification algorithms, respectively.

## 2 Background

Here, we review tree-based speculative decoding methods. In particular, we discuss the way existing methods choose the speculated tree structure (Section 2.1) and the algorithms they use to sample and verify the token trees (Section 2.2).

## 2.1 Tree construction

The primary tree structure used by existing methods is one composed of $k$ independent sequences of length $L$ that branch from the tree root (which corresponds to the current prefix). The SpecTr paper additionally considers arbitrary branching patterns $(k_1, k_2, ..., k_t)$, but says that this did not perform better in their experiments than independent sequences. Medusa constructs a full $k$-ary tree, which increases the success rate at each layer but cannot form a deep tree under moderate token budgets [4].

## 2.2 Tree sampling and verification

We now review how SpecInfer [28], SpecTr [40], naive sampling [28], and top-$k$ sampling[2] perform token tree sampling and verification. With regard to sampling, SpecInfer, SpecTr, and naive sampling all perform i.i.d. sampling with replacement from the draft model, while top-$k$ sampling selects the top-$k$ highest probability tokens from the draft model. In terms of verification, SpecInfer and SpecTr compare the draft and target model probabilities for the sampled tokens to decide which (if any) to accept; naive and top-$k$ sampling, on the other hand, sample a token from the *target model* distribution and accept it if it corresponds to one of the tokens from the speculated tree. These methods all verify a speculated token tree in a recursive manner—starting at the root of the tree—differing only in the verification algorithm they apply at each node.

**SpecInfer:** The SpecInfer method iteratively verifies tokens that were sampled from one or more draft models. Like the original speculative decoding method [24], it compares the draft model probabilities to those from the target model to decide if to accept. Note that while the SpecInfer method allows sampling from $k$ different draft models to generate $k$ children for a node, in this work we consider the more common setting where only one draft model is available. Therefore, we compare with the version of SpecInfer which samples from a single draft model $k$ times instead. To see pseudocode for SpecInfer, please see Algorithm 2 and ignore all blue lines (lines 10-16).

**SpecTr:** The SpecTr algorithm is similar in spirit to the SpecInfer algorithm. It iterates through the children of a node, and uses a sampling procedure to decide if to accept a child, in such a way that the output distribution is unchanged. One important property of this algorithm is that it is within a factor of $(1 - 1/e)$ of the best possible verification algorithm (i.e., the one with highest possible acceptance rate). For brevity, we refer readers to Algorithm 3 in the SpecTr paper for the exact pseudocode for this algorithm.

**Naive sampling and top-$k$ sampling:** Given a node in a token tree, the verification algorithm for naive sampling and top-$k$ sampling first samples from the *target model's* distribution $\mathcal{P}(\cdot \mid x_{<n})$ at that node, and then accepts this sample if it is equal to one of the children of that node. This verification algorithm trivially maintains the target model output distribution—regardless of how the token tree was generated—given that one always samples from the target model in this algorithm (as opposed to from the draft model, like in SpecTr and SpecInfer). This observation motivates our choice—for the top-$k$ sampling method—to populate the tree by taking the top-$k$ children of each node, instead of the naive sampling approach of taking $k$ i.i.d. samples (with replacement). We use the top-$k$ sampling method in our experiments in Section 3.2, to better understand the limits of this verification algorithm.

# 3 SEQUOIA

We now present SEQUOIA, a scalable and robust speculative decoding algorithm.

- In Section 3.1, we present our scalable tree construction algorithm, which uses dynamic programming to solve for the optimal tree structure. We demonstrate both theoretically and empirically that the number of tokens generated by verifying SEQUOIA trees scales nearly logarithmically in the size of the tree, while existing tree structures asymptote in the number of tokens they can generate.

- In Section 3.2, we present our robust tree verification algorithm, which modifies the SpecInfer algorithm by sampling *without replacement* from the draft model. We show both theoretically and empirically that SEQUOIA is robust, performing well across temperature values, while existing verification methods are not.

## 3.1 Tree construction

We now present the SEQUOIA tree construction algorithm (Section 3.1.1), and prove that the expected number of tokens generated when verifying for these trees scales well with the tree size (Section 3.1.2).

---

[2]Top-$k$ sampling is an improved version of naive sampling which we introduce as a baseline in this work.

### 3.1.1 Algorithm

To derive the SEQUOIA tree construction algorithm, we first express the tree construction problem as a constrained optimization problem, and then use dynamic programming to solve this problem optimally and efficiently. In this optimization problem, we aim to maximize the expected number of tokens $F(\mathcal{T})$ generated by verifying a token tree $\mathcal{T}$, under a constraint on the size of $\mathcal{T}$. We begin by presenting a closed form expression for $F(\mathcal{T})$ (Proposition 3.4). We then present our tree construction algorithm, which uses dynamic programming to find the tree of size $n$ which maximizes this expression (for any value of the speculation budget $n$).

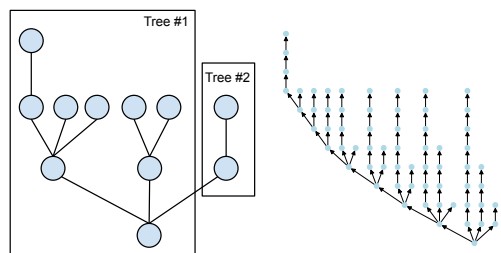

Figure 2: **Left**: Recursive sub-structure use by the dynamic programming algorithm. **Right**: Real example of SEQUOIA tree of size 64, and maximum depth 12. We present more examples of SEQUOIA trees in Figure 5 in Appendix E.

We first present a number of important definitions:

**Definition 3.1.** Under the *positional acceptance assumption*, the probability of a verification algorithm accepting a token $t$ which is the $k^{th}$ child of an already accepted token depends only on the value of $k$.

**Definition 3.2.** The *acceptance vector* is the vector $p = (p_1, p_2, ..., p_k, ...)$ containing the probabilities $p_k$ that the verification algorithm accepts a token at child position $k$. Under the positional acceptance assumption, the acceptance dynamics of a verification algorithm can be completely described by the acceptance vector.

**Definition 3.3.** Given an acceptance vector $p$ and a tree $\mathcal{T}$, we define the *score function* $f(v)$ for a node $v \in \mathcal{T}$ as $f(v) = \prod_{i \in \texttt{Path}(v)} p_i$. where $\texttt{Path}(v)$ is equal to the list of child indices along the path from the root to a node $v \in \mathcal{T}$. For example, if $v$ is the $3^{rd}$ child of the root's $2^{nd}$ child, then $\texttt{Path}(v) = [2, 3]$. We define $f(root) = 1$.

We are now ready to present Proposition 3.4 (proof in Appendix F.1.2), which shows the closed form equation for the expected number of tokens generated by verifying a token tree $\mathcal{T}$, under the positional acceptance assumption. This is the equation which our SEQUOIA dynamic program will optimize.

**Proposition 3.4.** *Let $\mathcal{T}$ be a token tree that is verified with the positional acceptance assumption, and let $f(v)$ denote the score function for a node $v \in \mathcal{T}$. Then the the expected number of tokens $F(\mathcal{T})$ generated by verifying $\mathcal{T}$ equals*

$$F(\mathcal{T}) = \sum_{v \in \mathcal{T}} f(v).$$

**SEQUOIA Dynamic Programing Algorithm.** The SEQUOIA tree construction algorithm finds the tree $\mathcal{T}$ of size $N$ which maximizes $F(\mathcal{T})$, using dynamic programming. Our algorithm works by iteratively filling in the following 2-dimension tensor $T$:

$$T(n, b) = \max_{\mathcal{T}, |\mathcal{T}| = n, \text{ FirstBranch}(\mathcal{T}) = b} F(\mathcal{T}), \quad \forall\, 0 \le n \le N, 0 \le b \le B. \tag{1}$$

Here, FirstBranch$(\mathcal{T})$ denotes the number of direct children the root of $\mathcal{T}$ has, and $B$ denotes an upper bound we impose on the number of direct children any node in the tree can have (we can let $B = N - 1$ to make this constraint vacuous). Given the tensor $T$, the maximum expected number of generated tokens for any tree of size $n \le N$ can be found by searching over all possible first-branch values $b$: $\max_{0 \le b \le B} T[n, b]$.

We now show how to iteratively fill in the tensor $T$ (which we initialize to negative $\infty$). Pseudocode for the full dynamic programming method is shown in Algorithm 1).

As the base case, we set $T[1, 0] = 1$, representing the tree composed of just the root node, because 1 token is generated per iteration of speculative decoding when no tokens are speculated.

For the recursive case, we can consider the tree composed of the root node and its first $b - 1$ children and their descendants (tree #1), as well as the tree whose root is the last child of the root node and

its descendants (tree #2). Letting $m \geq 1$ denote the number of nodes in tree #2, we can see that the expected number of generated tokens for tree #1 is $T[n-m, b-1]$. Furthermore, the expected number of generated tokens for tree #2 is $\max_{0 \leq j \leq B} T[m, \ j]$, but this sub-tree is only considered in the case where the $b^{th}$ child of the primary root node is accepted (which happens with probability $P[b]$). Therefore, we can compute $T[n, b]$ by searching over all possible sizes $m$ for tree #2 to find the one which maximizes the expected number of generated tokens for the full tree:

$$T[n, b] = \max_{1 \leq m \leq n-1} \left( T[n-m, b-1] + P[b] \cdot \max_{0 \leq j \leq B} T[m, j]\right).$$

We show in Appendix F.1.1 that by keeping track of the values of $m$ and $b$ that maximize the $\max$ expressions on lines 9 and 11, we can easily reconstruct the optimal tree $\mathcal{T}$ of size $N$ (and FirstBranch($\mathcal{T}) \leq B$) that attains the maximum expected number of generated tokens. We additionally demonstrate in this appendix (with python implementation) that we can extend this algorithm in a couple important ways:

- **Bounded tree-depth**: Because the amount of time it takes to speculate a token tree is proportional to the depth of the tree, it can be very beneficial to find the tree of depth $\leq D$ that maximizes the expected number of generated tokens. We demonstrate in Algorithm 4 that we can extend the SEQUOIA dynamic program to find the optimal tree of bounded depth.

- **Compatibility with self-speculation**: For self-speculation methods like Medusa [4], Eagle [25], and GLIDE [11] which leverage the target model's representations on the current prefix during decoding, the acceptance rates can meaningfully degrade as you get deeper into the speculation tree (i.e., further away from the current prefix). We demonstrate in Algorithm 4 that it is simple to extend our SEQUOIA dynamic program to take as input a 2-D acceptance rate *matrix* (instead of a 1-D vector) containing the average acceptance rate vectors at different tree depths. Thus, SEQUOIA is compatible with the latest advances in self-speculation methods, which can attain meaningfully higher acceptance rates than "standalone" draft models.

This algorithm can be run *offline*, and thus does not slow down inference.

---

**Algorithm 1** SEQUOIA Dynamic program

---

1: **Input:** $N$ for the maximum tree size, $B$ for the maximum number of branches of any node. $P[1], P[2], ..., P[B]$ for the probability of acceptance for each branch.
2: **Output:** $T[n, b]$ $\quad \forall \, 0 \leq n \leq N, 0 \leq b \leq B$.
3: Initialize array $T$, of size $(N+1, B+1)$, with $-\infty$ in all entries.
4: Initialize array $T_{max}$, of size $(N+1)$, with $-\infty$ in all entries.
5: $T[1, 0] = 1$
6: $T_{max}[1] = 1$
7: **for** $n = 2 \to N$ **do**
8: $\quad$ **for** $b = 1 \to B$ **do**
9: $\quad\quad$ $T[n, b] = \max_{1 \leq m \leq n-1} \left( T[n-m, b-1] + P[b] \cdot T_{max}[m]\right)$
10: $\quad$ **end for**
11: $\quad$ $T_{max}[n] = \max_{0 \leq b \leq B} T[n, b]$
12: **end for**
13: **Return array** $T$

---

### 3.1.2 Theoretical Results

We now prove that the SEQUOIA tree construction algorithm scales well with the size of the speculated tree. In particular, we show that under certain assumptions on the acceptance rates of the verification algorithm, the number of generated tokens is lower-bounded by a function which is (roughly) logarithmic in the size of the tree. This is in contrast to existing tree construction algorithms, which are upper bounded in the expected number of tokens they generate, regardless of the size of the tree. For example, a single sequence of tokens has upper bound $1/(1-P_1)$ [24]; $k$ independent sequences can only increase this upper bound by 1, because they only increase the chance of acceptance of the first token. Even an infinitely deep binary tree is upper bounded by $1/(1-P_2)$.

We first define what it means for a verification algorithm to have a $b$ *power-law acceptance rate*, and then present our theorem on the scalability of SEQUOIA trees, under the assumption that the verification algorithm has a $b$ power-law acceptance rate.

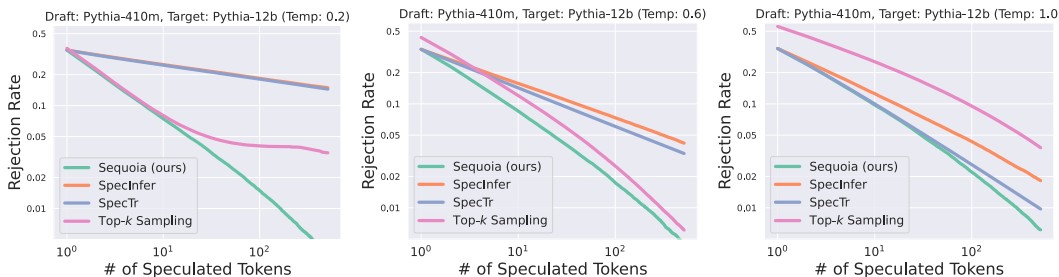

Figure 3: **Rejection rate vs. number speculated tokens**: We plot the average rejection rate $(1 - acceptance\_rate)$ for the different verification algorithms, as a function of the number of speculated tokens $k$. Across temperature settings ($\{0.2, 0.6, 1.0\}$, left to right), the SEQUOIA verification algorithm attains the lowest rejection rates, and consistently has a *power-law acceptance rate* (Definition 3.5).

**Definition 3.5.** We say that a tree verification algorithm has a $b$ *power-law acceptance rate* if the chance $r_k$ of the tree verification algorithm rejecting all $k$ speculated children of a node in a tree is upper bounded by a power-law of $k$ with exponent $b$—meaning, $r_k \leq 1/k^b \; \forall k \in \mathbb{N}$, for $b > 0 \in \mathbb{R}$.

The above definition is motivated by our observation (Figure 3) that the SEQUOIA sampling/verification algorithm attains power-law acceptance rates in practice. We now state the theorem (proof in App. F.3).

**Theorem 3.6.** *Using a tree verification algorithm with a $b$ power-law acceptance rate, the expected number of tokens $G(n)$ generated by verifying the SEQUOIA tree of size $n$ is in $\Omega\big(b \cdot \log(n)/\log(\log(n))\big)$.*

### 3.1.3 Empirical Validation

In Figure 1, we plot the average number of tokens generated by SEQUOIA trees relative to various baseline tree structures, as a function of the number of tokens $n$ in the tree, using Pythia-2.8B as a draft model for Pythia-12B, and WikiText-103. We see that the number of generated tokens for SEQUOIA trees is unbounded—scaling roughly logarithmically with the tree size—whereas the other tree structures asymptote. We show results for more draft/target model pairs in Figure 6 in Appendix G.3.

## 3.2 Tree sampling and verification

We present our token tree sampling and verification algorithm, and prove it is the first such algorithm to satisfy two important robustness properties, while maintaining the target model's output distribution.

### 3.2.1 Algorithm

We present the pseudocode for the SEQUOIA Tree sampling and verification algorithm in Algorithm 2. As discussed in Section 2, an important motivation for designing the SEQUOIA verification algorithm was the observation that SpecInfer and SpecTr both perform poorly at low temperatures, due to the fact that they can repeatedly sample (and then reject) a low-quality token that the draft model is confident in. Thus, we wanted to design an algorithm that would never make the same mistake twice—meaning, once a token was rejected, it would never propose that token again. Toward this end, SEQUOIA introduces two changes to the SpecInfer algorithm (shown in blue text in Algorithm 2): First, it performs sampling *without replacement* using the draft model distribution. Second, if all the tokens with non-zero draft model probability have already been sampled and rejected, it uses the uniform distribution over all tokens that have not yet been sampled as the new draft model distribution. These changes significantly improve the robustness of SEQUOIA relative to SpecInfer, while maintaining the guarantee that the output distribution is identical to that of the target model (proof in Appendix F.2.1).

### 3.2.2 Theoretical Results

We now prove that the SEQUOIA verification algorithm is robust, in the sense that it satisfies both of the properties below, while existing verification algorithms do not.

**Algorithm 2** SEQUOIA Sampling and Verification
*(The blue lines [10-16] distinguish SEQUOIA's sampling/verification from SpecInfer's [28])*

1: **Input:** Prefix $[x_1, x_2, ..., x_{n-1}]$, target model probabilities $\mathcal{P}(\cdot\,|\,x_{<n})$, draft model probabilities $\mathcal{Q}(\cdot\,|\,x_{<n})$, and number of branches $k \leq vocab\_size$.
2: **Output:** A token $x$ sampled using SEQUOIA.
3: Initialize residual $R$ with $\mathcal{P}$, draft $D$ with $\mathcal{Q}$, and the set of rejected tokens $S$ with $\varnothing$
4: **for** $i = 1 \rightarrow k$ **do**
5:     sample $s_i \sim D$, $r_i \sim \mathrm{Uniform}(0,1)$
6:     **if** $r_i < \frac{R[s_i]}{D[s_i]}$ **then**
7:         **Return** $s_i$       # Accept $s_i$
8:     **else**
9:         $R \leftarrow \mathrm{norm}(\max(R - D, 0))$
10:         $D[s_i] \leftarrow 0$
11:         $S.\mathrm{add}(s_i)$
12:         **if** $\mathrm{sum}(D) = 0$ **then**
13:             # Let $D$ be uniform over non-rejected set
14:             $D[t] \leftarrow 0$ if $t \in S$, else 1
15:         **end if**
16:         $D \leftarrow \mathrm{norm}(D)$
17:     **end if**
18: **end for**
19: **Return** $x \sim R$

---

- **The *optimal transport* property**: When $k = 1$, the acceptance rate is equal to $1 - \frac{\|P-Q\|_1}{2}$.[3]
- **The *cover* property**: If the support of the draft model probability distribution $Q$ is of size $k$ and is a superset of the support of the target model probability distribution $P$, at most $k$ speculations will be needed to attain an acceptance rate of 1. Furthermore, if $k$ is equal to the vocabulary size, the acceptance rate should always be 1 as well, regardless of the draft model used.

Intuitively, satisfying the *optimal transport* property results in strong performance at high temperatures (because $P$ and $Q$ will approach uniform distributions), while satisfying the *cover* property results in strong performance at low temperatures (if top target model token is in the top-$k$ draft model tokens).

We now present our main robustness result (proof in Appendix F.3):

**Theorem 3.7.** SEQUOIA *verification satisfies both properties (optimal transport, cover); SpecInfer & SpecTr only satisfy the optimal transport property; top-$k$ sampling only satisfies the cover property.*

### 3.2.3 Empirical Validation

In Figure 3, we plot the average rejection rates (equal to $1 -$ acceptance rates) for the different verification algorithms, as a function of the number of speculated child tokens for a fixed token prefix, for various temperatures (0.2, 0.6, 1.0), measured on WikiText-103. We can see that across all temperature settings, the rejection rates for SEQUOIA decay faster than for the other algorithms. In general, we observe that the rejection rates $r_k$ for SEQUOIA follow a power-law, where $r_k \approx 1/k^b$ for some $b > 0$. We can also see that while SpecTr and SpecInfer perform relatively well at high temperatures, they struggle at lower temperatures, and that the opposite is true for top-$k$ sampling.

## 4 Evaluation

In this section, we aim to demonstrate that SEQUOIA can speed up LLM inference by a large margin in wall-clock time. We first present our end-to-end system results showing total speedup, followed by validating our claims that SEQUOIA is scalable and robust.

- In Section 4.1, we demonstrate SEQUOIA's superior end-to-end performance. Specifically, SEQUOIA achieves up-to $4.04\times$ speed-up for Llama2-7B on A100 and $9.5\times$ for Llama3-70B on L40 offloading (achieving the latency as low as 0.60 s/token).
- In Section 4.2.1, we show that the SEQUOIA tree can generate on average 33% more tokens than a tree of 16 independent sequences (tree size 512).

---

[3] The SpecTr paper [40] showed that $1 - \frac{\|P-Q\|_1}{2}$ is the acceptance rate attained by the optimal verification algorithm for $k = 1$.

Table 1: **On-device results (A100)**: The optimal tree configuration and speedup for different pairs of draft and target models, and different temperatures, for SEQUOIA vs. SpecInfer. We specify the average number of generated tokens per decoding step in parentheses, next to the speedup factor. SEQUOIA attains up to $4.04\times$ speedup on an A100. The speed of incremental decoding is **24.2ms/token** with Huggingface. The draft model speed is 0.5ms/token. TBT refers to time between tokens.

| Target LLM | Draft Model | T | Dataset | Tree Config. (size, depth) | Speedup | TBT ms/token | SpecInfer $5\times 8$ | Speedup vs SpecInfer |
|---|---|---|---|---|---|---|---|---|
| Llama2-7B | JF68M | 0 | C4 | (128,10) | **4.04 ×(5.08)** | 6.0 | 3.45×(3.96) | 1.17× |
| Llama2-7B | JF68M | 0.6 | C4 | (128,7) | **3.18×(3.92)** | 7.6 | 2.47×(2.97) | 1.29× |
| Llama2-7B | JF68M | 0 | OpenWebText | (128,7) | **3.22×(3.86)** | 7.5 | 2.79×(3.15) | 1.15× |
| Llama2-7B | JF68M | 0.6 | OpenWebText | (128,6) | **2.71×(3.33)** | 8.9 | 2.10×(2.54) | 1.29× |
| Llama2-7B | JF68M | 0 | CNN Daily | (128,7) | **3.41×(4.05)** | 7.1 | 2.95×(3.27) | 1.16× |
| Llama2-7B | JF68M | 0.6 | CNN Daily | (128,6) | **2.83×(3.45)** | 8.5 | 2.11×(2.58) | 1.34× |
| Llama2-7B | JF68M | 0 | MT Bench | (128,10) | **4.03×(4.98)** | 6.0 | 3.84×(4.01) | 1.05× |
| Llama2-7B | JF68M | 0.6 | MT Bench | (128,7) | **3.18×(3.96)** | 7.6 | 2.45×(2.97) | 1.30× |

Table 2: **Offloading results (L40)**: The optimal tree configuration and speedup for different pairs of draft and target models, and different temperatures, for SEQUOIA vs. SpecInfer. We specify the average number of generated tokens per decoding step in parentheses, next to the speedup factor. SEQUOIA attains up to $9.5\times$ speedup in the offloading setting on an L40. The speed of incremental decoding is **5.7s/token** with DeepSpeed Zero Inference. TBT refers to time between tokens.

| Target LLM | Draft Model | T | Dataset | Tree Config. (size, depth) | Speedup | TBT s/token | SpecInfer $16\times 48$ | Speedup vs SpecInfer |
|---|---|---|---|---|---|---|---|---|
| Llama2-70B-chat | Llama2-7B-chat | 0 | MT Bench | (768,18) | **8.6×(10.30)** | 0.66 | 5.7×(7.63) | 1.51× |
| Llama2-70B-chat | Llama2-7B-chat | 0.6 | MT Bench | (768,18) | **8.4×(9.91)** | 0.68 | 5.2×(7.03) | 1.62× |
| Llama3-70B-Instruct | Llama3-8B-Instruct | 0 | MT Bench | (768,18) | **9.5×(11.68)** | 0.60 | 7.0×(9.07) | 1.36× |
| Llama3-70B-Instruct | Llama3-8B-Instruct | 0.6 | MT Bench | (768,18) | **9.3×(11.37)** | 0.61 | 6.1×(8.29) | 1.52× |

- In Section 4.2.2, show SEQUOIA's sampling and verification algorithm is robust to temperature, consistently outperforming SpecInfer (by up to $1.65\times$) and top-$k$ sampling (by up to $1.27\times$).

## 4.1 End-to-end Results

We now demonstrate that SEQUOIA speeds up LLM decoding in the on-device setting by up $4.04\times$ on an A100 GPU, and up to $9.5\times$ with offloading on an L40 GPU.

**Setup.** Our experiments are based on Llama and Vicuna models. For the on-device setting, we use JackFram/Llama-68m (JF68m) [28] and princeton-nlp/Sheared-Llama-1.3B (SL1.3B) [46] as the draft models, and Llama2-7B [43], Llama2-13B, and Vicuna-33B [6] as the target models. For the offloading setting, we use Llama2-7B-chat/Llama3-8B-Instruct as the draft model and Llama2-70B-chat/Llama3-70B-Instruct as the target model. We evaluate our results on C4(en) [35] validation dataset, OpenWebText [14], CNN DailyMail [36] and MT Bench [52]. In each experiment, we use 200 examples to measure the acceptance rate vector (mentioned in Section 3.1) and sample another 200 examples for evaluation (50 for offloading). The prompt length and generation length are both set to 128 tokens except MT Bench. We evaluate SEQUOIA on different hardware including on-device experiments on L40 and A100(-PCIE 80GB) GPUs, as well as offloading experiments on an L40 GPU (with PCIE4). We also compare SEQUOIA with SpecInfer [28] with $5\times 8$ trees (5 independent sequences of length 8, the tree structure used in [28] for batch size 1) for the on-device setting, and $16\times 48$ trees for the offloading setting.

**Implementation Details.** We implement the draft and target models using Transformers [45]. Because we determine the optimal tree structure in advance, we are able to use PyTorch CUDA graphs [31, 32] to reduce the overhead of kernel launching during speculative decoding. To accelerate sampling without replacement—which is not efficient in PyTorch 2.1 [32]—we use the exponential-sort algorithm [44], combined with PyTorch CUDA graphs [31, 32]. For offloading setting, we used an DeepSpeed-Zero-Inference [2] as baseline, which is 5.7 s/token.

**Hardware-Aware Optimization.** For each hardware setting we consider in our experiments, we use the following method for selecting the size and depth of the Sequoia tree we should use to maximize speedups, while avoiding doing an exhaustive grid search. Letting $G(n,d)$ denote the expected number of tokens generated by verifying the SEQUOIA tree of size $n$ and depth $d$ (computed via dynamic programming), $t(n)$ denote the (hardware-dependent) amount of time it takes the target model to verify

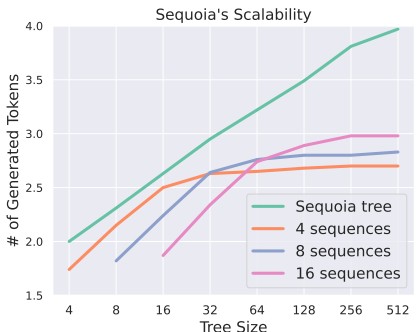
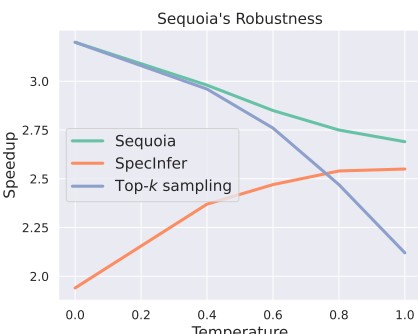

Figure 4: **Left**: We compare the number of tokens generated on average by SEQUOIA trees vs. $k$ independent sequences, where we use SEQUOIA sampling and verification for both tree structures. **Right**: We compare the speedups attained by the SEQUOIA sampling and verification algorithm relative to SpecInfer and top-$k$ sampling, across various temperatures, holding the tree structure fixed.

$n$ tokens divided by the time to verify 1 token, and $c$ denote the (hardware-dependent) time to draft 1 token divided by the time to verify 1 token, the speedup attained by SEQUOIA can be expressed as $\text{Speedup}(n,d) = \frac{G(n,d)}{t(n)+d\cdot c}$. We measure $t(n)$ and $c$ empirically for each type of model and inference hardware, and then search over possible values of $n$, $d$ to find the pair that gives the largest speedup.

**Main Results.** We evaluate SEQUOIA using different temperatures, draft and target model pairs, and hardware configurations. Results are shown in Table 1 (A100 on-device) and Table 2 (L40 offloading). We observe that SEQUOIA consistently speeds up LLM decoding in a wide range of settings. SEQUOIA reaches up to $4.04\times$ speedup for the on-device setting, and up to $9.5\times$ speedup for the offloading setting, as a result of the huge gap between computation capacity and memory bandwidth. Notably, for the offloading setting on L40, SEQUOIA can achieve as low as 0.60 s/token latency. We present additional on-device results (A100 and L40) in Appendix G.

**Analysis.** We made several interesting observations on the interplay between SEQUOIA tree construction, sampling and verification, and hardware-aware optimizer. (1) SEQUOIA selects much larger trees in the offloading setting (768 tokens) than in the on-device setting (64 to 128 tokens). (2) In general, the average number of generated tokens is close to the wall-clock time speedup (especially when JF68M is used as the draft) as a result of the hardware-aware tree optimizer. (3) The optimal trees found by SEQUOIA for slightly different configurations—e.g., different temperatures and model pairs—can be very different from one another. (4) SEQUOIA chooses deeper trees at low temperature than high temperature, due to the acceptance rates being higher for low temperature.

## 4.2 Ablations

We present our ablation experiments validating the scalability of the SEQUOIA tree construction algorithm (Section 4.2.1), and the robustness of SEQUOIA tree sampling and verification algorithm (Section 4.2.2). For each of these experiments, we only vary one element at a time (e.g., the tree structure for Section 4.2.1) to study the gains attained by each component of SEQUOIA.

### 4.2.1 The Scalability of SEQUOIA

In Figure 4 (left) we compare the average number of generated tokens for the SEQUOIA tree construction method, relative to $k$ independent sequences, at different budgets; we use SEQUOIA's sampling and verification algorithm for all trees. The SEQUOIA tree is able to generate up to 33% more tokens per decoding step, demonstrating the effectiveness of SEQUOIA's tree construction algorithm. Here, we use JackFram/Llama-68m as the draft model, Llama2-13B as the target model, $0.6$ as the temperature, and CNN Daily Mail as the dataset.

### 4.2.2 Robustness of SEQUOIA Sampling Algorithm

In Figure 4 (right) we compare the SEQUOIA sampling and verification algorithm to SpecInfer and top-$k$ sampling across different temperature values, holding the tree structure fixed. We can see that

SEQUOIA achieves the largest speedups across all temperatures, attaining up to $1.65\times$ and $1.27\times$ speedup relative to SpecInfer and top-$k$ sampling, respectively. Here, we use JackFram/Llama-68m as the draft model, Llama2-7B as the target model, CNN Daily Mail as the dataset, and the corresponding SEQUOIA tree from Table 1 (temperature $0.6$) as the tree structure. In Table 8 in Appendix G.4, we additionally show that the SEQUOIA sampling/verification algorithm is robust to the top-$p$ parameter.

## 5 Conclusion

We presented SEQUOIA, a scalable and robust speculative decoding method. By improving the topology of the token tree and the sampling algorithms, SEQUOIA is able to speed up autoregressive LLM inference up to $4.04\times$ on GPU and $9.5\times$ with offloading. In addition to providing real speedups, we believe SEQUOIA also provides insight into both the large potential and fundamental limits of speculative decoding systems. We hope that this understanding inspires future work in this area, or even informs the design of custom chips for LLM inference.

## Acknowledgments

We thank Xinyu Yang, Harry Dong, Ranajoy Sadhukhan, Hanshi Sun, Silong Yong and the anonymous reviewers for their helpful discussions and feedback on the paper. This work was partially supported by the National Science Foundation under grant numbers CNS-2147909, CNS-2211882, and CNS-2239351, along with gift awards from Amazon, Cisco, Google, Intel, Li Auto, Meta, Moffet AI, Oracle, Qualcomm, and Samsung.

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

## A   Broader Impacts

In this paper, we present a new algorithm for accelerating speculative decoding. While there are numerous application scenarios of large language models that warrant additional study regarding possible societal impact, we would like to highlight that our work does not advance the capabilities of these models. Our work is primarily an algorithmic study with no specific usage limitations, and while LLMs themselves can be used with malicious purpose, we believe that none of such use cases are specific to this paper.

## B   Limitations

**Theoretical limitations:**   On the theoretical front, there are two primary limitations to our results:

1. **The positional acceptance assumption (Definition 3.1**: The optimality of our dynamic program depends on this assumption. In particular, this assumption states that the *only* factor influencing the acceptance rate for a token is what "number child" it is to it's "parent token" (e.g., if it is the first or fifth sampled token to follow the "parent" token). This allows us to model the acceptance dynamics using simple closed form equations, which ignore all contextual factors impacting acceptance rates (e.g., the current prefix, the confidence of the draft model, etc.).

2. **The $b$ power law acceptance rate (Definition 3.5)**: While we observe in our experiments that SEQUOIA satisfies this assumption (see Figure 3), it's important to note that need this assumption for our theoretical results on the scalability of SEQUOIA trees to hold (Theorem F.2).

**Methodological limitations:**   In terms of the limitations of SEQUOIA in practice, the most important limitation/challenge is likely that the structure of the optimal SEQUOIA tree depends on the exact (average) acceptance rate vector, which depends on the draft/target model pair, temperature value, data domain, etc. The optimal tree also depends on the batch size, which can be considered by the hardware-aware optimizer. It is relatively work-intensive to have to measure the acceptance rate vector for each setting, and use this vector to compute the optimal tree. In practice, we believe computing a single tree for a typical use case can work well for other use cases (e.g., higher/lower temperatures, different data domains), but we leave a more thorough analysis of this issue for future work.

## C   Related Work

This work introduces a new algorithm in the family of speculative decoding methods that aims to maintain the exact output distribution of the target model by improving the structure and sampling/verification algorithm for the speculated token tree. There exist many other directions within this line of work—for example, methods which introduce leniency into the speculative decoding algorithm to attain increased speed at the cost of accuracy [22, 38], methods that reuse layers or representations from the target model as the draft model [50, 4], etc. Alternatively, the draft model can be distilled to better approximate the target model; DistillSpec [53, 18, 41, 42] improves that process by using model-generated data and adjusting the objective depending on the task and the decoding strategy. Finally, LLMCad [47] proposes an advanced algorithm for token tree generation and verification in the context of on-device LLM inference.

In addition to speculative decoding, there exist many other methods aimed at improving the speed of LLM inference. For example, model quantization is another very promising way of dealing with the I/O bottleneck during inference, by reducing the number of bits per parameter. However, unlike speculative decoding, these methods generally deteriorate the quality of the model to some degree, depending on the amount of quantization [17, 20, 30, 51, 26, 13, 10] or sparsity [29, 27, 19].

Meanwhile, various works [12, 39, 48, 1] have studied ways to improve LLM serving throughput. Pope et al. [33] investigated the batching effect in scaling up LLM. Orca [49] proposed a distributed LLM serving system that uses a finegrained scheduling policy to improve GPU utilization under various request lengths. vLLM [23] used page tables to manage GPU memory to increase memory utilization, which significantly boosts inference throughput. FlexGen [37] proposed an offloading mechanism to support larger batches to achieve high throughput.

FlashAttention [9, 8] is another algorithm that aims to improve the speed of LLMs (at both training and inference time) by considering the I/O cost of different operations.

Another promising approaching to speeding up inference is to change the fundamental building blocks of the model. Recently, numerous sub-quadratic architectures—including SSMs [16, 15] and linear attention models [21]—have been proposed. These models are particularly beneficial for long inputs.

## D   Background: Sequence-based speculative decoding

The original speculative decoding method [24, 5] proposes using a small "draft model" to speculate $\gamma$ tokens into the future, and then using the "target model" to in parallel process these tokens and decide which of the tokens to "accept", in such a way that the output distribution of the target model is unchanged. This algorithm is presented in Algorithm 3.

Leviathan et al. [24] analyze the performance of this algorithm, presenting equations for the expected number of accepted tokens from one run of the algorithm, and the expected wall-clock speed up from using speculative decoding (relative to standard autoregressive inference with the target model). In this analysis, they introduce the acceptance rate $\alpha \in [0,1]$, corresponding to the probability that a token $x_i$ is accepted by Algorithm 3, under the simplifying assumption that the acceptance decisions are i.i.d.[4] Under this assumption, they show that the expected number of generated tokens in each run of Algorithm 3 is $\frac{1-\alpha^{\gamma+1}}{1-\alpha}$. Additionally, letting $c$ denote the ratio between the time to run the draft model and the time to run the target model, they show that the expected wall-clock speed-up from using this algorithm is $\frac{1-\alpha^{\gamma+1}}{(1-\alpha)(\gamma c+1)}$.

---

**Algorithm 3** Sequence-based Speculative Decoding

---

1: **Input:** Prefix $[x_1, x_2, ..., x_{n-1}]$, Target model $M_p$, draft model $M_q$, and number of tokens $\gamma$ to speculate.
2: **Output:** A sequence of tokens generated using speculative decoding.
3: **for** $i = n \rightarrow n+\gamma$ - 1 **do**                    ▷ Sample sequence of $\gamma$ tokens from draft model
4:     $q_i(x) \leftarrow M_q([x_1,...,x_{i-1}])$
5:     $x_i \sim q_i(x)$
6: **end for**
7: **for** $i = n \rightarrow n+\gamma$ **do**      ▷ For loop below can be run in parallel with a single forward pass of $M_p$
8:     $p_i(x) \leftarrow M_q([x_1,...,x_{i-1}])$
9: **end for**
10: $s \leftarrow n-1$                                    ▷ Choose how many tokens $n$ to accept
11: **for** $i = n \rightarrow n+\gamma$ - 1 **do**
12:     $r_i \sim \text{Uniform}(0,1)$
13:     **if** $r_i < \frac{p_i(x_i)}{q_i(x_i)}$ **then**
14:         $s \leftarrow s+1$
15:     **else**
16:         break
17:     **end if**
18: **end for**
19: $p'(x) \leftarrow p_{s+1}(x)$
20: **if** $t < n+\gamma-1$ **then**
21:     $p'(x) \leftarrow \text{norm}(\max(0, p_{s+1}(x) - q_{s+1}(x)))$
22: **end if**
23: $t \sim p'(x)$                                    ▷ Sample a final token from $p'(x)$
24: **Return** $x_1,...,x_s, t$

---

---

[4]One can think of $\alpha$ as the average acceptance rate over many runs of this algorithm on a representative dataset.

# E  Examples of SEQUOIA trees

Below we show more examples of SEQUOIA trees of various sizes. Note that for these plots we do not limit the depth of the tree. The acceptance rate vector we used for this (shown below) was computed with Llama3-70B-Instruct target model, Llama3-8B-Instruct draft model, on CNN daily news dataset: [0.7732, 0.1039, 0.0402, 0.0206, 0.0128, 0.0081, 0.0064, 0.0043, 0.0035, 0.0026, 0.0025, 0.0021, 0.0016, 0.0014, 0.0010, 0.0010, 0.0010, 0.0007, 0.0007, 0.0006, 0.0007, 0.0006, 0.0004, 0.0004, 0.0005, 0.0006, 0.0004, 0.0003, 0.0002, 0.0004, 0.0001].

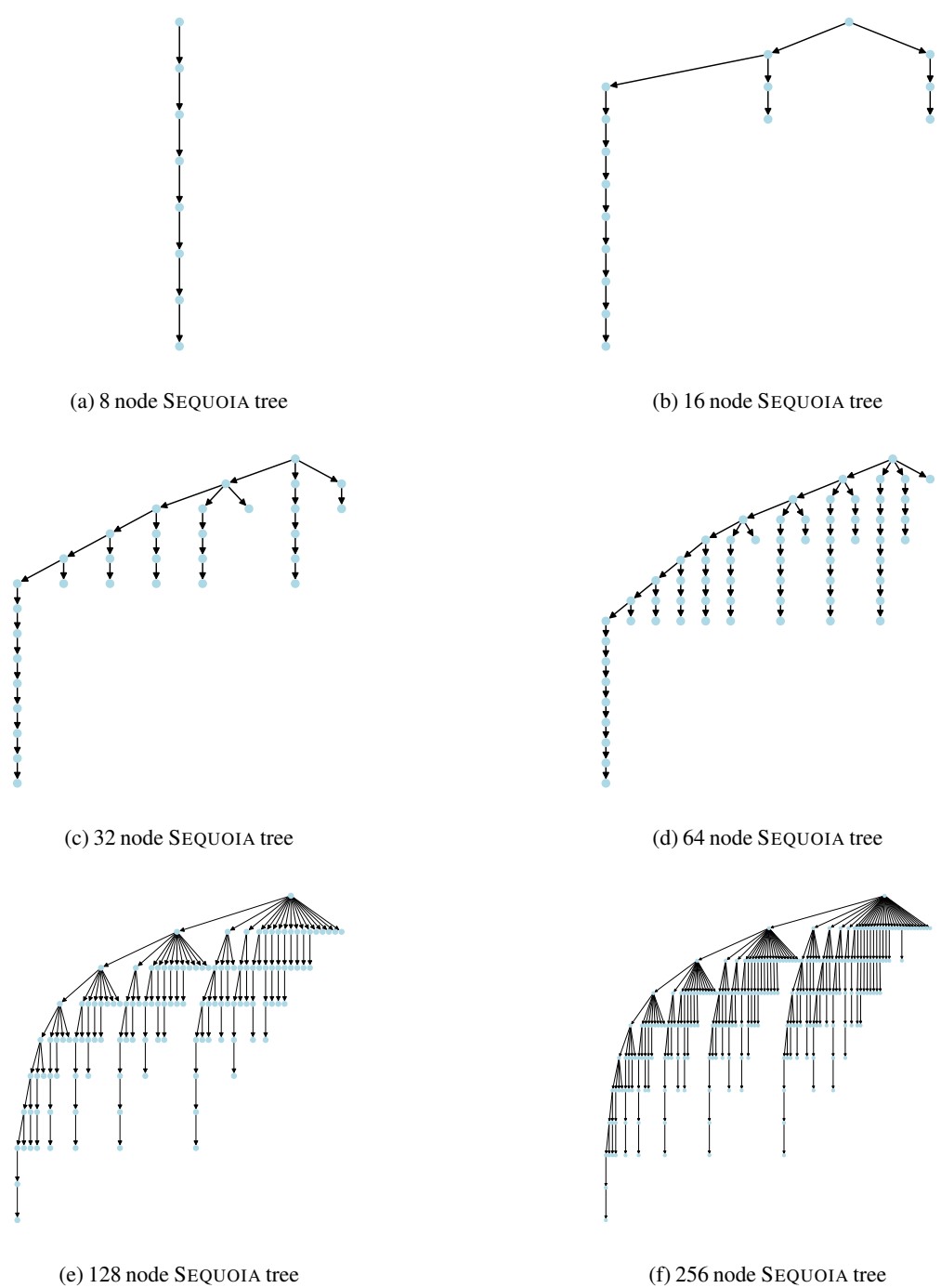

(a) 8 node SEQUOIA tree

(b) 16 node SEQUOIA tree

(c) 32 node SEQUOIA tree

(d) 64 node SEQUOIA tree

(e) 128 node SEQUOIA tree

(f) 256 node SEQUOIA tree

Figure 5: A set of increasingly large SEQUOIA trees.

# F   Method details and theoretical results

We present additional details (as well as proofs for theorems) about the SEQUOIA tree construction (Section F.1) and tree sampling and verification (Section F.2) methods.

## F.1   SEQUOIA tree construction algorithm

We begin by presenting details about the SEQUOIA tree construction algorithm, and its corresponding theoretical properties.

### F.1.1   SEQUOIA dynamic program details

In this section, we present an extended version of the SEQUOIA tree construction dynamic programming (DP) algorithm (Algorithm 1), including a full python implementation of this extended algorithm (Algorithm 4). In Algorithm 1, we showed how to compute the expected number of generated tokens for the optimal tree of size $N$ (and branching factor $\leq B$). Here, we extend the algorithm to be able to handle:

1. An upper bound $D$ on the depth of the token tree, and
2. Self-speculation methods like Eagle [25] whose acceptance rates decay for tokens that are deeper in the speculated tree.

We then show how to additionally generate the optimal *tree structure* using dynamic programming, for these more general settings.

**Extensions to bounded depth and self-speculation methods:**   To handle the above cases, we assume that we have a 2-D array $p$, where $P[d,\ b]$ is the probability of acceptance for a node at depth $d$ and branch number $b$. Here we assume $p$ is zero-indexed, so depth $0$ corresponds to the direct children of the root node. We also assume $p$ has shape $(D-1, B+1)$, where $D$ is the limit on the depth of the speculated tree, and $B$ is the limit on the branch factor of the tree (max number of children per node). This allows us to infer that when we are computing $T[n,d,b]$ (during the internal running of the DP algorithm), in the case where the root node has a depth limit of $D$, the node being considered has depth limit $d$ it must be at depth $D-d$; thus, $D-d$ is the index of the $P$ array (at dimension 0) that should be used at that time. Using this fact, we can show that the recursion equation for Eagle (with bounded depth) is quite similar the one from Equation 1 (and Algorithm 1) in Section 3.1.1:

$$T[n,d,b] = \max_{1 \leq m \leq n-1} \left( T[n-m,d,b-1] + P[D-d,b] \cdot \max_{0 \leq j \leq B} T[m,d-1,j] \right)$$
$$\forall\, 2 \leq n \leq N,\ 2 \leq d \leq D,\ 2 \leq b \leq B.$$

**Constructing the optimal tree structure:**   In the python implementation below of the extended SEQUOIA DP algorithm (Algorithm 4), we show how to recursively construct the optimal tree structure for each tree size $n$ and depth limit $d$. Throughout the DP we maintain the following data structures:

- $best\_new\_node[n,d,b]$: A pointer to the root of the best sub-tree to add as the $b^{th}$ child of the tree root with budget n, depth <= d, and b children.
- $best\_tree[n,d]$: A pointer to root of the best tree with n nodes and depth <= d.

Line 32, and then lines 40-45, demonstrate the recursive relationship between these tree structures:

- If $m^*$ is the optimal number of tokens that should be assigned to the tree rooted at the $b^{th}$ (and last) child for the tree (For the $(n,d,b)$ tree), then we can look up the optimal tree of that size in $best\_tree[m^*,d-1]$, and set $best\_new\_node[n,d,b] = best\_tree[m^*,d-1]$.
- If $b^*$ is the optimal number of children for a tree of size $n$ and depth $\leq d$, we can look-up $best\_new\_node[n,\ d,\ b^*]$ (the root of a tree of size $m'$) and assign that as the last child of $best\_tree[n,d]$. To then find the optimal $(b-1)^{th}$ child of this tree, we can look-up $best\_new\_node[n-m',d,b^*-1]$, and we can continue in this manner until we have added all $b^*$ children to $best\_tree[n,d]$.

This demonstrates how to build the optimal tree as part of the dynamic program.

**Algorithm 4** SEQUOIA tree construction algorithm: Python implementation

```python
import numpy as np

class Node:
    def __init__(self, children=None):
        self.children = children if children is not None else []
        self.num_nodes_in_tree = 1 + sum(c.num_nodes_in_tree for c in self.children)

def sequoia_tree_construction(acc_rates, max_tree_size, max_tree_depth, max_branch):
    P, N, D, B = acc_rates, max_tree_size, max_tree_depth, max_branch
    if P.ndim == 1:
        P = np.tile(P, (D - 1, 1))
    assert P.shape == (D - 1, B + 1)

    T = np.full(shape=(N + 1, D + 1, B + 1), fill_value=-float('inf'))
    T_max = np.full(shape=(N + 1, D + 1), fill_value=-float('inf'))
    T[1, 1:, 0] = 1.0
    T_max[1, 1:] = 1.0

    # best_new_node[n, d, b] = A pointer to the best node (tree root node) to add
    #     as the b^th child of the tree root with budget n, depth <= d, and b children.
    # best_tree[n, d] = A pointer to root of the best tree with n nodes and depth <= d.
    best_new_node = {(1, d, 0): None for d in range(1, D + 1)}
    best_tree = {(1, d): Node() for d in range(1, D + 1)}

    for n in range(2, N + 1):
        for d in range(2, D + 1):
            for b in range(1, B + 1):
                x = np.nan_to_num(T[n - 1: 0: -1, d, b - 1] + P[D - d, b] * T_max[1: n, d - 1],
                                  nan=0.0, neginf=-float('inf'))
                T[n, d, b] = np.max(x)
                if T[n, d, b] > 0.0:
                    best_new_node[n, d, b] = best_tree[np.argmax(x) + 1, d - 1]
            T_max[n, d] = np.max(T[n, d, :])

            if T_max[n, d] > 0:
                best_b = np.argmax(T[n, d, :])
                best_n_budget_depth_d_tree_children = []
                remaining_budget = n
                # Find the `best_b` children of the root node, starting with the last.
                for b in range(best_b, 0, -1):
                    next_child = best_new_node[remaining_budget, d, b]
                    best_n_budget_depth_d_tree_children.insert(0, next_child)
                    remaining_budget -= next_child.num_nodes_in_tree
                assert remaining_budget == 1
                best_tree[n, d] = Node(children=best_n_budget_depth_d_tree_children)

    return T, best_tree
```

### F.1.2  Proof of Proposition 3.4: Closed-form expression for $F(\mathcal{T})$

We now prove Proposition 3.4 by deriving the closed-form expression for $F(\mathcal{T})$ (the expected number of tokens generated by verifying tree $\mathcal{T}$), and show how to use dynamic programming to find the optimal tree $\mathcal{T}$ under a tree budget size.

**Proposition F.1.** *Let $\mathcal{T}$ be a token tree that is verified with the positional acceptance assumption, and let $f(v)$ denote the score function for a node $v \in \mathcal{T}$. Then the expected number of tokens $F(\mathcal{T})$ generated by verifying $\mathcal{T}$ is equal to*

$$F(\mathcal{T}) = \sum_{v \in \mathcal{T}} f(v).$$

*Proof.* Let $D(\mathcal{T})$ denote the expected number of tokens generated by verifying tree $\mathcal{T}$. We would like to prove that $D(\mathcal{T}) = F(\mathcal{T}) \forall \mathcal{T}$. We will prove this by induction on the size of $\mathcal{T}$.

**Base case** ($N = 1$): A tree of size 1 is composed solely of the root node. By definition of the score function $f(v)$ (Definition 3.3), we know that $f(v) = 1$ for the root node, so $F(\mathcal{T}) = 1$. $D(\mathcal{T}) = 1$ also, because verifying a tree composed of a root node with no children will simply sample from the target model, and generate 1 token.

**Inductive step** ($N > 1$): For $|\mathcal{T}| = N > 1$, let $v$ be a leaf of $\mathcal{T}$ at child index $i_v$ of depth $d$ with parent $v_p$ and sibling $\mathcal{S}_v$ (set of sibling indices). We can then consider the tree $\mathcal{T}' = \mathcal{T} - \{v\}$. Based on the inductive assumption, we know that $g(\mathcal{T}') = D(\mathcal{T}')$. Using this assumption, we can express $D(\mathcal{T})$ in terms of $D(\mathcal{T}')$:

$$D(\mathcal{T}) = D(\mathcal{T}') - (d-1) \cdot f(v_p) \cdot \left(1 - \sum_{i \in \mathcal{S}_v} p_i\right) + (d-1) \cdot f(v_p) \cdot \left(1 - \sum_{i \in \mathcal{S}_v \cup \{i_v\}} p_i\right) + d \cdot f(v)$$

$$= D(\mathcal{T}') - (d-1)f(v_p)p_{i_v} + d \cdot f(v)$$

$$= \sum_{v' \in \mathcal{T}'} f(v') - (d-1)f(v) + d \cdot f(v)$$

$$= F(\mathcal{T}') + f(v)$$

$$= F(\mathcal{T})$$

Note that we use the inductive hypothesis, along with the fact the $f(v_p) \cdot p_{i_v} = f(v)$ (by definition of $f(v)$). $\qquad\qquad\square$

### F.1.3 Proof of Theorem 3.6: Main scalability results for SEQUOIA trees

We now prove that, under certain assumptions on the acceptance rates of the tree verification algorithm, the expected number of tokens generated by verifying the SEQUOIA tree is lower bounded by a function which is roughly logarithmic in the size of the tree. We will do this by showing that a simpler tree—the $k^*(n)$ tree (defined below)—also has this lower bound, and using the fact that the SEQUOIA tree is by construction the tree with the largest expected number of generated tokens.

We define the $k^*(n)$ tree to be the $k$-ary tree[5] with $\leq n$ nodes that has the highest expected accepted sequence length. Letting $G(n)$ denote the expected accepted sequence length for the $k^*(n)$ tree, we will now prove that $G(n) \in \Omega\big(b\log(n)/\log(\log(n))\big)$ (meaning, it is lower-bounded by a scalar multiple of $b\log(n)/\log(\log(n))$), under the assumption that the rejection rate $r_k$ is upper-bounded by a power-law of $k$. It then follows directly (as a corollary) that the growth rate of the tree generated by the SEQUOIA algorithm will also be in $\Omega\big(b\log(n)/\log(\log(n))\big)$.

**Theorem F.2.** *Assume the chance $r_k$ of a token tree verification algorithm rejecting all $k$ speculated tokens ($k$ child nodes of some node in the tree) is upper bounded by a power-law of $k$; so $r_k \leq 1/k^b$ for some $b > 0 \in \mathbb{R}$. Then the growth rate $G(n)$ for the $k^*(n)$ tree is in $\Omega\big(b\log(n)/\log(\log(n))\big)$.*

*Proof.* We will let $k(n) = \lfloor \log(n)^{1/b} \rfloor$ denote the branch-width chosen for tree size $n$, and show that under this assumption, the growth rate $G'(n)$ of the corresponding $k(n)$-tree is at least $\frac{b\log(n)}{10\log(\log(n))}$, assuming that $n$ is large enough. Given that $G'(n)$ is a lower bound on $G(n)$ (because the above choice of $k(n)$ might not be fully optimal), and using the definition of $\Omega$, this proves that $G(n) \in \Omega\big(b\log(n)/\log(\log(n))\big)$. Note that we will abbreviate $k(n)$ as $k$ in many places throughout the proof, for brevity.

---

[5] Recall that a $k$-ary tree is one where every non-leaf node has $k$ children.

If we let $d$ denote the depth of the tree, the number of nodes in the tree is $1 + k + k^2 + \ldots + k^d = \frac{k^{d+1}-1}{k-1} \leq n$. This implies $d \leq \log_k(n)$, which we can prove as follows:

$$k^{d+1} - 1 \leq n(k-1)$$
$$\Rightarrow k^{d+1} \leq nk - n + 1 \leq nk$$
$$\Rightarrow d+1 \leq \log_k(nk) = \log_k(n) + 1$$
$$\Rightarrow d \leq \log_k(n)$$

We can assume $d$ is the largest integer such that $d \leq \log_k(n)$, so it also follows that $d+1 \geq \log_k(n)$.

Letting $\alpha_k \coloneqq 1 - r_k$, the expected length $G'(n)$ of the accepted token sequence can be expressed as $1 \cdot (1-\alpha_k) + 2\alpha_k \cdot (1-\alpha_k) + 3\alpha_k^2(1-\alpha_k) + \ldots + (d+1)\alpha_k^d = 1 + \alpha_k + \alpha_k^2 + \ldots + \alpha_k^d = \frac{1-\alpha_k^{d+1}}{1-\alpha_k}$ (the first equality is a result of telescoping sums, the second is from the sum of a finite geometric series). We will now lower bound this expression, making use of Lemma F.4 (defined and proven below).

$$
\begin{aligned}
G(n) \geq G'(n) &= \frac{1-\alpha_k^{d+1}}{1-\alpha_k} \\
&= \frac{1-(1-r_k)^{d+1}}{r_k} \\
&\geq \frac{d+1}{10} \quad \text{applying Lemma F.4, and assuming } r_k \cdot (d+1) \leq \log(1.9) \\
&\geq \frac{\log_k(n)}{10} \\
&= \frac{\log(n)}{10\log(k)} \\
&\leq \frac{\log(n)}{10\log(\log(n)^{1/b})} \\
&= \frac{b\log(n)}{10\log(\log(n))}
\end{aligned}
$$

Now we simply need to understand when $r_k \cdot (d+1) \leq \log(1.9)$:

$$
\begin{aligned}
r_k \cdot (d+1) &\leq \frac{1}{k^b}\left(\log_k(n) + 1\right) \\
&\leq \frac{2\log_k(n)}{(\log(n)^{1/b}-1)^b} \quad \text{using } k(n) = \lfloor \log(n)^{1/b} \rfloor \geq \log(n)^{1/b} - 1 \\
&\leq \frac{2\log_k(n)}{(\frac{1}{2}\log(n)^{1/b})^b} \quad \text{assuming } \log(n)^{1/b} \geq 2 \Leftrightarrow n \geq \exp(2^b) \\
&= \frac{2^{b+1}\log(n)}{\log(k)\log(n)} \\
&= \frac{2^{b+1}}{\log(k)}
\end{aligned}
$$

So if $\frac{2^{b+1}}{\log(k)} \leq \log(1.9)$, then it follows that $r_k \cdot (d+1) \leq \log(1.9)$.

$$\frac{2^{b+1}}{\log(k)} \leq \log(1.9) \Leftrightarrow \frac{2^{b+1}}{\log(1.9)} \leq \log(k) \Leftrightarrow \exp\left(\frac{2^{b+1}}{\log(1.9)}\right) \leq k$$

Given that $k(n) = \lfloor \log(n)^{1/b} \rfloor \geq \log(n)^{1/b} - 1$, we know that if $\log(n)^{1/b} - 1 \geq \exp\left(\frac{2^{b+1}}{\log(1.9)}\right)$, then

it must hold that $k(n) \geq \exp\left(\frac{2^{b+1}}{\log(1.9)}\right)$ as well. We can see that this holds if:

$$\log(n)^{1/b} - 1 \geq \exp\left(\frac{2^{b+1}}{\log(1.9)}\right) \Leftrightarrow n \geq \exp\left(\left(1 + \exp\left(\frac{2^{b+1}}{\log(1.9)}\right)\right)^b\right)$$

Thus, we have shown that as long as $n$ is greater than the above expression, then $G'(n) \geq \frac{b\log(n)}{10\log(\log(n))}$. Because we know that $G(n) \geq G'(n)$, this concludes the proof that $G(n)$ is in $\Omega\big(b\log(n)/\log(\log(n))\big)$.
$\square$

We now prove, as a corollary of Theorem F.2, that the growth rate of the SEQUOIA tree is also in $\Omega\big(b\log(n)/\log(\log(n))\big)$.

**Corollary F.3.** *Under the same assumptions on the rejection rates as Theorem F.2, it holds that the growth rate for the* SEQUOIA *tree is in* $\Omega\big(b\log(n)/\log(\log(n))\big)$.

*Proof.* By construction, for every tree size $n$, the SEQUOIA tree is the tree that has the largest expected number of generated tokens. Thus, for every value of $n$ the expected number of generated tokens for the SEQUOIA tree must be larger than that of the $k^*(n)$ tree, which was shown in Theorem F.2 to be in $\Omega\big(b\log(n)/\log(\log(n))\big)$. This concludes the proof.
$\square$

We now prove the lemma that we used to prove Theorem F.2:

**Lemma F.4.** *For any real number* $x \in (0,1]$, *and integer* $m > 0$ *such that* $mx \leq \log(1.9)$, *it holds that* $\frac{1-(1-x)^m}{x} \geq \frac{m}{10}$.

*Proof.*

$$\frac{1-(1-x)^m}{x} = \frac{1 - \left(1 - mx + \binom{m}{2}x^2 - \binom{m}{3}x^3 + \binom{m}{4}x^4 - \ldots + (-1)^m x^m\right)}{x}$$

$$= \frac{mx - \binom{m}{2}x^2 + \binom{m}{3}x^3 - \binom{m}{4}x^4 + \ldots - (-1)^m x^m\Big)}{x}$$

$$= m - \binom{m}{2}x + \binom{m}{3}x^2 - \binom{m}{4}x^3 + \ldots - (-1)^m x^{m-1}$$

$$\geq m - \binom{m}{2}x - \binom{m}{3}x^2 - \binom{m}{4}x^3 - \ldots - x^{m-1}$$

$$\geq m - \frac{m^2}{2!}x - \frac{m^3}{3!}x^2 - \frac{m^4}{4!}x^3 - \ldots$$

$$= m\left(1 - \frac{mx}{2!} - \frac{(mx)^2}{3!} - \frac{(mx)^3}{4!} - \frac{(mx)^4}{5!} - \frac{(mx)^5}{6!} - \ldots\right)$$

$$= m\left(2 - 1 - \frac{mx}{2!} - \frac{(mx)^2}{3!} - \frac{(mx)^3}{4!} - \frac{(mx)^4}{5!} - \frac{(mx)^5}{6!} - \ldots\right)$$

$$= m\left(2 - \left(1 + \frac{mx}{2!} + \frac{(mx)^2}{3!} + \frac{(mx)^3}{4!} + \frac{(mx)^4}{5!} + \frac{(mx)^5}{6!} - \ldots\right)\right)$$

$$\geq m\left(2 - \left(1 + mx + \frac{(mx)^2}{2!} + \frac{(mx)^3}{3!} + \frac{(mx)^4}{4!} + \frac{(mx)^5}{5!} + \ldots\right)\right)$$

$$= m\left(2 - e^{mx}\right)$$

$$\geq \frac{m}{10} \qquad \text{Assuming } e^{mx} \leq 1.9, \text{ which is true by our initial assumption.}$$

$\square$

### F.2 SEQUOIA sampling and verification algorithm

We now move on to presenting proofs about the correctness and robustness of the SEQUOIA sampling and verification method.

#### F.2.1 Proof of correctness for the SEQUOIA sampling and verification algorithm

We prove now that the SEQUOIA verification algorithm maintains the output distribution of the target model. We assume we have a target model $t$, and a list of draft models $(d_1,...d_n,d_{n+1},...)$, where $d_i$ in this case depends on the previously rejected samples $x_1,...,x_{i-1}$, and where $d_i(u)$ and $t(u)$ denote the probabilities of sampling token $u \in V$ from $d_i$ or $t$ respectively (where $V$ is the token vocabulary). We let $t_i$ denote the residual at iteration $i$ of SEQUOIA loop, (after $i-1$ nodes have been rejected (so $t_1 = t$, as can be seen in Algorithm 2)

We will prove by induction on the number of proposed tokens $n$ that the SEQUOIA verification algorithm is correct.

**Base case** ($n = 0$): SEQUOIA is trivially correct, as it will simply sample from the residual $t_1$, which is equal to $t$.

**Recursive case**: We assume SEQUOIA is correct for $n-1$ proposed samples and prove it is correct for $n$ proposed samples.

We first show that at stage $i$ in the speculative decoding algorithm, the chance of SEQUOIA choosing to reject the proposed sample is equal to $\sum_x \max\left(0, t_i(x) - d_i(x)\right)$:

**Lemma F.5.** *P(No token accepted at iteration i)* $= \sum_x \max\left(0, t_i(x) - d_i(x)\right)$.

*Proof.*

$$P(\text{No token accepted at iteration } i) = \sum_x P(\text{sample } x) \cdot P(\text{reject } x \mid x \text{ is sampled})$$

$$= \sum_x d_i(x) \cdot \left(1 - \min\left(\frac{t_i(x)}{d_i(x)}, 1\right)\right)$$

$$= \sum_x d_i(x) - \sum_x \min\left(t_i(x), d_i(x)\right)$$

$$= \sum_x t_i(x) - \sum_x \min\left(t_i(x), d_i(x)\right)$$

$$= \sum_x t_i(x) + \max\left(-t_i(x), -d_i(x)\right)$$

$$= \sum_x t_i(x) - t_i(x) + \max\left(0, t_i(x) - d_i(x)\right)$$

$$= \sum_x \max\left(0, t_i(x) - d_i(x)\right)$$

$\square$

We are now ready to prove the recursive case of the SEQUOIA algorithm. By the inductive hypothesis, we know that for all $u \in V$,

$$t(u) = P(u \text{ accepted in first } n-1 \text{ iterations}) + P(\text{No token accepted in first } n-1 \text{ iterations}) \cdot t_n(u)$$

What this means is that in the case where we run SEQUOIA for $n-1$ iterations (and if no token is accepted we sample from the residual $t_n$), this is equivalent to sampling from the target distribution $t$ directly. We would like to show that this output distribution is equivalent to the one we would get if we run SEQUOIA for $n$ iterations (and if no token is accepted we sample from the residual $t_{n+1}$). The

output distribution of this scenario can be written as follows:

$P(u$ accepted in first $n-1$ iterations$)+P($No token accepted in first $n-1$ iterations$)\cdot$

$$\left( d_n(u)\cdot P(u \text{ accepted at iteration } n)+P(\text{No token accepted in iteration } n)\cdot t_{n+1}(u) \right)$$

Thus, all we must show is that

$$t_n(u)=d_n(u)\cdot P(u \text{ accepted at iteration } n)+P(\text{No token accepted in iteration } n)\cdot t_{n+1}(u)$$

We now show this desired result. We will use Lemma F.5, and the fact that by definition of the SpecInfer algorithm (see Algorithm 2, ignoring blue lines), we know that $t_{n+1}(u)=\dfrac{\max\left(0, t_n(u)-d_n(u)\right)}{\sum_x\max\left(0, t_n(x)-d_n(x)\right)}$.

$$d_n(u)\cdot P(u \text{ accepted at iteration } n)+P(\text{No token accepted in iteration } n)\cdot t_{n+1}(u)$$

$$=d_n(u)\cdot\min\left(1,\frac{t_n(u)}{d_n(u)}\right)+\left(\sum_x\max\left(0,t_n(x)-d_n(x)\right)\right)t_{n+1}(u)$$

$$=\min\left(d_n(u),t_n(u)\right)+\left(\sum_x\max\left(0,t_n(x)-d_n(x)\right)\right)\cdot\left(\frac{\max\left(0,t_n(u)-d_n(u)\right)}{\sum_x\max\left(0,t_n(x)-d_n(x)\right)}\right)$$

$$=\min\left(d_n(u),t_n(u)\right)+\max\left(0,t_n(u)-d_n(u)\right)$$

$$=t_n(u)$$

To see that this last equality holds, we consider two cases:

1. Case 1 $\left(t_n(u)\geq d_n(u)\right)$: $\min(d_n(u),t_n(u))+\max(0,t_n(u)-d_n(u))=d_n(u)+t_n(u)-d_n(u)=t_n(u)$.

2. Case 1 $\left(t_n(u)<d_n(u)\right)$: $\min(d_n(u),t_n(u))+\max(0,t_n(u)-d_n(u))=t_n(u)+0=t_n(u)$.

This completes the proof.

### F.3 Proof of Theorem 3.7: Main robustness result for SEQUOIA sampling and verification

We now prove the robustness results for the SEQUOIA verification algorithm.

**Theorem F.6.** *The* SEQUOIA *verification algorithm satisfies both the optimal transport and the cover properties, while SpecInfer and SpecTr only satisfy the optimal transport property, and (top-$k$) naive sampling only satisfies the cover property.*

*Proof.* This proof is quite straightforward:

- **SEQUOIA satisfies the optimal transport property**: It is clear that SEQUOIA satisfies the optimal transport property, because at $k=1$, it is identical to the original speculative decoding algorithm [24].

- **SEQUOIA satisfies the cover property**: To see why SEQUOIA satisfies the cover property, we will use the following two facts:
  - If the support of $Q$ is of size $k$ and $k$ tokens are speculated by the draft model, the set of speculated tokens will always exactly equal the $k$ tokens in the support of $Q$ (because SEQUOIA does sampling without replacement from the draft model).
  - During the verification for-loop in Algorithm 2, the support of the residual will always be contained in the support of $P$ intersected with the set of tokens that have not yet been rejected. This is because the support of the residual can never grow (because $p_i(x) = 0 \Rightarrow p_{i+1}(x) = norm(max(p_i - q_i, 0))(x) = 0$, where $p_i$ and $q_i$ denote the residual and draft probabilities at iteration $i$, respectively), and because if a token $x$ is rejected it will "exit" the residual (because $x$ is rejected implies $q_i(x) > p_i(x)$ which implies that $p_{i+1}(x)=norm(max(p_i-q_i,0))(x)=0$).

Combining these two facts, we can see that if the first $k-1$ tokens were rejected, then the $k^{th}$ token must be accepted, because the residual must be a one-hot vector with probability 1 at the only remaining token, and the (updated) draft probabilities will also be this same one-hot vector (and thus, accepted with probability 1). Additionally, we can see that if $V$ tokens are sampled (where $V$ is the vocab size), these must exactly equal the $V$ tokens in the vocabulary, and thus one of those tokens must be accepted. In the case where the support of $Q$ is equal to the full vocabulary, this result follows directly from the discussion above. In the case where the support of $Q$ *does not* equal the full vocabulary, this is a result of the fact that once all tokens in the support of $Q$ have been sampled and rejected, we begin sampling (without replacement) from the uniform distribution over all non-rejected tokens.

- **SpecInfer satisfies the optimal transport property**: For $k=1$, SpecInfer is identical to the original speculative decoding algorithm [24].

- **SpecInfer does not satisfy the cover property**: It is easy to see that SpecInfer does not satisfy the cover property, with the following counter-example. Let $Q=[0.5,0.5]$ and $P=[1.0,0]$. We can see that the support of $Q$ is of size 2 and contains the support of $P$. But with probability 25%, SpecInfer will sample the second token twice in a row, and will reject both of them.

- **SpecTr satisfies the optimal transport property**: For $k=1$, SpecTr is identical to the original speculative decoding algorithm [24], because $\gamma=1$ by definition.

- **SpecTr does not satisfies the cover property**: We can show that SpecTr (in particular, the '$k$-sequential selection' algorithm from [40]) does not satisfy the cover property, with the following counter-example. Let $P=[1,0]$ and $Q=[0.5,0.5]$. Then $\beta_{p,q}(\gamma) = \sum_{x=0}^{1} \min(Q(x),P(x)/\gamma) = \min(0.5,1/\gamma) + \min(0.5,0/\gamma) = 0.5$ (because $\gamma \in [1,2]$ by assumption). We know the acceptance rate of SpecTr is $1-(1-\beta_{p,q}(\gamma))^2 = 1-(1-0.5)^2 = 0.75 \neq 1$. Thus, SpecTr does not satisfy the cover property.

- **Top-$k$ naive sampling does not satisfy the optimal transport property**: Letting $Q=[0.6,0.4]$ and $P=[0.6,0.4]$, we can see that top-$k$ naive sampling will accept with probability 0.6, whereas $1-\|P-Q\|/2 = 1.0$.

- **Top-$k$ naive sampling satisfies the cover property**: It's easy to see that if the support of $Q$ is of size $k$ and contains the support of $P$, then top-$k$ naive sampling will always accept (because it will sample from the target model and accept if the sampled token is among the top-$k$ tokens according to the draft model). Similarly, if $k=V$, it must accept as well (because the top-$V$ tokens must be the full vocabulary, and so any sample from the target model must accept).

$\square$

# G    Additional Experiments

## G.1    Additional end-to-end speedup results

We provide additional end-to-end results comparing SEQUOIA to baselines, extending the results from Section 4.1. Here (Tables 3 and 4), we provide on-device results on A100 and L40 GPUs, for a more extended set of models, relative to the results in Table 1, but on different hardware.

## G.2    More Comparisons with SpecInfer

To demonstrate the optimality of SEQUOIA's tree construction, we provide a sweep of tree configurations and corresponding speedups of SpecInfer in Tables 5 and 6. SEQUOIA attains better speedups in both greedy decoding and stochastic decoding than all tree configurations of SpecInfer.

## G.3    Scalability Additional Results

Here we present additional results demonstrating the scalability of the SEQUOIA tree construction algorithm relative to baselines, for several Pythia draft and target model pairs on the WikiText-103 dataset:

Table 3: **On-device results (A100)**: The optimal tree configuration and speedup for different pairs of draft and target models, and different temperatures, for SEQUOIA vs. SpecInfer. We specify the average number of generated tokens per decoding step in parentheses, next to the speedup factor. SEQUOIA attains up to $4.04\times$ speedup on an A100. TBT refers to time between tokens.

| Target LLM | Draft Model | T | Dataset | Tree Config. (size, depth) | Speedup | TBT ms/token | SpecInfer $5\times8$ |
|---|---|---|---|---|---|---|---|
| Llama2-7B | JF68M | 0 | C4 | (128,10) | **4.04** ×**(5.08)** | 6.0 | 3.45×(3.96) |
| Llama2-7B | JF68M | 0.6 | C4 | (128,7) | **3.18**×**(3.92)** | 7.6 | 2.47×(2.97) |
| Llama2-7B | JF68M | 0 | OpenWebText | (128,7) | **3.22**×**(3.86)** | 7.5 | 2.79×(3.15) |
| Llama2-7B | JF68M | 0.6 | OpenWebText | (128,6) | **2.71**×**(3.33)** | 8.9 | 2.10×(2.54) |
| Llama2-7B | JF68M | 0 | CNN Daily | (128,7) | **3.41**×**(4.05)** | 7.1 | 2.95×(3.27) |
| Llama2-7B | JF68M | 0.6 | CNN Daily | (128,6) | **2.83**×**(3.45)** | 8.5 | 2.11×(2.58) |
| Llama2-13B | JF68M | 0 | C4 | (64,9) | **3.73**×**(4.20)** | 8.4 | 3.30×(3.64) |
| Llama2-13B | JF68M | 0.6 | C4 | (64,7) | **3.19**×**(3.57)** | 9.8 | 2.48×(2.87) |
| Llama2-13B | JF68M | 0 | OpenWebText | (64,7) | **3.18**×**(3.49)** | 9.8 | 2.77×(3.05) |
| Llama2-13B | JF68M | 0.6 | OpenWebText | (64,6) | **2.77**×**(3.06)** | 11.3 | 2.17×(2.49) |
| Llama2-13B | JF68M | 0 | CNN Daily | (64,7) | **3.33**×**(3.68)** | 9.4 | 2.95×(3.22) |
| Llama2-13B | JF68M | 0.6 | CNN Daily | (64,6) | **2.88**×**(3.17)** | 10.8 | 2.17×(2.54) |
| Vicuna-33B | SL1.3B | 0 | C4 | (64,6) | **2.27**×**(4.28)** | 23.4 | 1.83×(3.86) |
| Vicuna-33B | SL1.3B | 0.6 | C4 | (64,6) | **2.19**×**(4.16)** | 24.3 | 1.64×(3.53) |
| Vicuna-33B | SL1.3B | 0 | OpenWebText | (64,5) | **2.21**×**(3.93)** | 24.1 | 1.75×(3.70) |
| Vicuna-33B | SL1.3B | 0.6 | OpenWebText | (64,5) | **2.13**×**(3.82)** | 25.0 | 1.57×(3.36) |
| Vicuna-33B | SL1.3B | 0 | CNN Daily | (64,5) | **2.21**×**(3.93)** | 24.1 | 1.75×(3.71) |
| Vicuna-33B | SL1.3B | 0.6 | CNN Daily | (64,5) | **2.16**×**(3.86)** | 24.6 | 1.58×(3.40) |

Table 4: **on-device results (L40)**: The optimal tree configuration and speedup for different pairs of draft and target models, and different temperatures, for SEQUOIA vs. SpecInfer. We specify the average number of generated tokens per decoding step in parentheses, next to the speedup factor. SEQUOIA attains up to $3.95\times$ speedup on an L40.

| Target LLM | Draft Model | T | Dataset | Tree Config. (size, depth) | Speedup | SpecInfer $5\times8$ |
|---|---|---|---|---|---|---|
| Llama2-7B | JF68M | 0 | C4 | (64,10) | **3.95**×**(4.68)** | 3.50×(3.98) |
| Llama2-7B | JF68M | 0.6 | C4 | (64,7) | **3.10**×**(3.63)** | 2.28×(2.89) |
| Llama2-7B | JF68M | 0 | OpenWebText | (64,7) | **3.12**×**(3.58)** | 2.79×(3.16) |
| Llama2-7B | JF68M | 0.6 | OpenWebText | (64,6) | **2.68**×**(3.12)** | 2.08×(2.54) |
| Llama2-7B | JF68M | 0 | CNN Daily | (64,7) | **3.30**×**(3.79)** | 2.89×(3.28) |
| Llama2-7B | JF68M | 0.6 | CNN Daily | (64,6) | **2.81**×**(3.27)** | 2.09×(2.59) |
| Llama2-13B | JF68M | 0 | C4 | (64,10) | **3.15**×**(4.25)** | 2.76×(3.61) |
| Llama2-13B | JF68M | 0.6 | C4 | (64,8) | **2.62**×**(3.57)** | 2.06× (2.81) |
| Llama2-13B | JF68M | 0 | OpenWebText | (64,8) | **2.64**×**(3.52)** | 2.34×(3.05) |
| Llama2-13B | JF68M | 0.6 | OpenWebText | (64,6) | **2.28**×**(3.07)** | 1.79×(2.44) |
| Llama2-13B | JF68M | 0 | CNN Daily | (64,7) | **2.78**×**(3.68)** | 2.47×(3.21) |
| Llama2-13B | JF68M | 0.6 | CNN Daily | (64,7) | **2.37**×**(3.22)** | 1.85×(2.51) |

### G.4   Robustness Additional Results

See Table 8.

### G.5   Evaluation of SEQUOIA hardware-aware optimizer

In this section, we demonstrate the effectiveness of the SEQUOIA hardware-aware tree optimizer. We compare the speedups attained by the SEQUOIA trees of various sizes from Figure 4 (left) to the trees selected by the hardware-aware tree-optimizer. Because the tree optimizer is able to limit the tree depth to make speculation faster, it is able to attain larger end-to-end speedups than any of the SEQUOIA trees from Figure 4 (left), whose structures were chosen to maximize the expected number of generated tokens (not the speedup). The optimizer is also able to automatically find the tree size that produces the largest overall speedup.

Table 5: A sweep of tree configurations and their corresponding speedups of SpecInfer [28] on A100. The draft model is JF68M, and the target model is Llama2-7B in greedy decoding. The evaluated dataset is C4. The default tree configuration in SpecInfer is $5\times8$, which brings $3.45\times$ speedup while SEQUOIA achieves $4.04\times$ speedup, surpassing all tree configurations below.

| Width/Depth | 1 | 2 | 4 | 8 | 16 | 32 | 64 | 128 |
|---|---|---|---|---|---|---|---|---|
| 1 | | | | 3.09× | 3.14× | 2.75× | 1.94× | 1.19× |
| 2 | | | 2.95× | 3.36× | 3.46× | 2.69× | 1.74× | |
| 4 | | 2.4× | 3.14× | 3.46× | 3.41× | 2.47× | | |
| 8 | 1.88× | 2.44× | 3.14× | 3.70× | 3.03× | | | |
| 16 | 2.00× | 2.55× | 3.27× | 3.14× | | | | |
| 32 | 1.86× | 2.57× | 2.81× | | | | | |
| 64 | 1.92× | 2.22× | | | | | | |
| 128 | 1.68× | | | | | | | |

Table 6: A sweep of tree configurations and their corresponding speedups of SpecInfer [28] on A100. The draft model is JF68M, and the target model is Llama2-7B in stochastic decoding. The evaluated dataset is C4. The default tree configuration in SpecInfer is $5\times8$, which brings $2.47\times$ speedup while SEQUOIA achieves $3.18\times$ speedup, surpassing all tree configurations below.

| Width/Depth | 1 | 2 | 4 | 8 | 16 | 32 | 64 | 128 |
|---|---|---|---|---|---|---|---|---|
| 1 | | | | 2.08× | 1.87× | 1.48× | 1.11× | 0.69× |
| 2 | | | 2.14× | 2.2× | 1.89× | 1.46× | 1.07× | |
| 4 | | 1.99× | 2.3× | 2.28× | 1.95× | 1.53× | | |
| 8 | 1.73× | 2.09× | 2.42× | 2.42× | 2.14× | | | |
| 16 | 1.78× | 2.07× | 2.41× | 2.18× | | | | |
| 32 | 1.78× | 2.08× | 2.24× | | | | | |
| 64 | 1.73× | 2.04× | | | | | | |
| 128 | 1.61× | | | | | | | |

Table 7: A sweep of tree configurations and their corresponding speedups of SpecInfer [28] on L40 offloading setting. The draft model is Llama2-7B-chat, and the target model is Llama2-70B-chat in stochastic decoding. The evaluated dataset is MT-Bench. SEQUOIA achieves $8.4\times$ speedup, surpassing all tree configurations below.

| Tree Config. | (16,48) | (24,32) | (32,24) |
|---|---|---|---|
| **Speedup** | 5.2× | 5.3× | 5.5× |

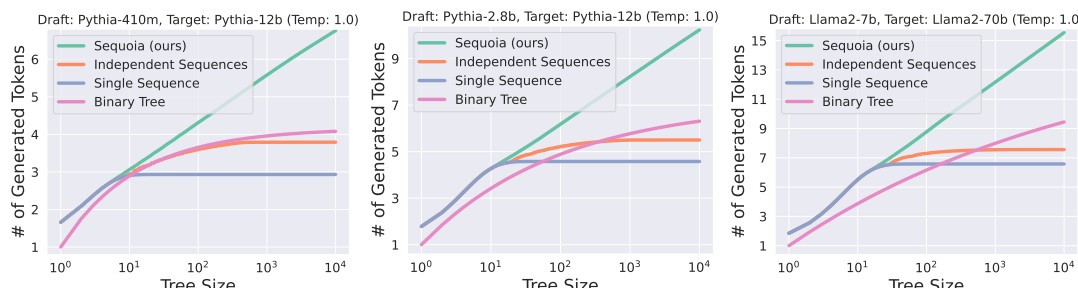

Figure 6: **Number of generated tokens vs. tree size**: We plot the average number of tokens generated for different tree structures per decoding step of the target model, as a function of the tree size, for different draft and target model pairs. The number of generated tokens for SEQUOIA trees continues to grow with the tree size, while other tree structures asymptote.

As mentioned in Section 4.1, one of the inputs to the hardware aware optimizer is $t(n)$, which is the hardware-dependent amount of time it takes the target model to verify $n$ tokens divided by the time to verify 1 token. In Figure 8 we show the forward pass times for different models on different hardware, for different number of tokens $n$. As you can see, the forward pass times are roughly constant for low values of $n$, but then eventually start growing roughly linearly in $n$—the value of $n$ at which $t(n)$ begins to grow is model and hardware dependent. In general, this value of $n$ is *lower* for hardware that has

Table 8: We compare the robustness of the SEQUOIA sampling and verification algorithm to the top-$p$ hyperparameter, relative to SpecInfer and top-$k$ sampling. We present total speedups on an A100 GPU for the different methods (number of generated tokens in parentheses). We hold the tree structure fixed across methods, use JF68M as the draft model, and Llama2-7B as the target model.

| Top-$p$ | SEQUOIA (Ours) | SpecInfer | top-$k$ sampling |
|---|---|---|---|
| 0.8 | 2.54×(3.18) | 2.35×(2.93) | 2.43×(2.90) |
| 0.9 | 2.61×(3.27) | 2.42×(3.01) | 2.27×(2.71) |
| 1.0 | 2.69×(3.26) | 2.55×(3.10) | 2.12×(2.44) |

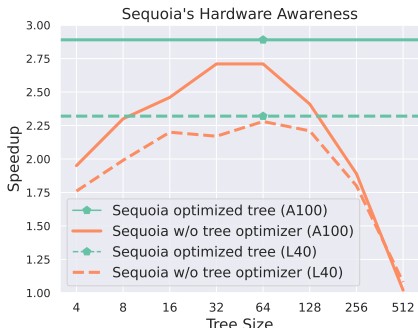

Figure 7: We compare the wall-clock time speedup of SEQUOIA trees of various sizes (orange lines)—chosen to maximize the # generated tokens—with the speedup of the trees selected by the hardware-aware tree optimizer (horizontal green lines)—chosen to maximize speedup—on A100 and L40 GPUs. The optimizer can select the optimal tree size and depth for each type of hardware; by limiting the depth of the tree it can make speculation faster and thus attain larger speedups than the trees with unconstrained depth (orange lines).

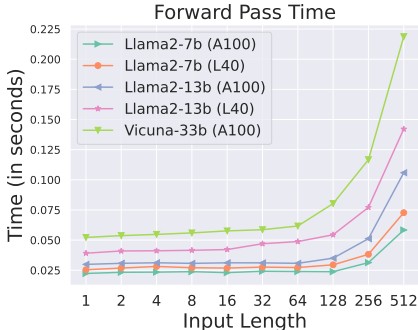

Figure 8: Forward pass times for different model/hardware combinations as a function of the number of tokens $n$ being processed. We use these values to choose the optimal tree.

a *higher* ratio of bandwidth (between GPU HBM and SRAM) to FLOPS, because it is less memory bound).

