# OpenReview forum: "Sequoia: Scalable and Robust Speculative Decoding"
_NeurIPS.cc/2024/Conference — NeurIPS 2024 spotlight_

### Official Review · Reviewer_yRtD · 2024-07-06

**Soundness:** 3
**Presentation:** 3
**Contribution:** 3
**Rating:** 6
**Confidence:** 4

**Summary:**

This paper introduces a speculative decoding method Sequoia, which uses a novel sampling and verification method that outperforms prior work across different decoding temperatures. The speedup of Sequoia is large.

**Strengths:**

1. This paper discuss the proposed method in a detailed way. The algorithm is novel and achieves large speedup.
2. This paper have a thorough and sound evaluation section. Sequoia outperforms SpecInfer in various datasets and temperature settings.

**Weaknesses:**

1.  Sequoia shows linear speedup with the tree size growing exponentially. However, it is less useful in batch serving (e.g., vllm). Do you think this problem can be solved by integrating a more efficient draft token proposing method (e.g., Medusa/Eagle) with the proposed Sequoia?

**Questions:**

I notice that the speedup of speculative decoding decreases as the temperature grows (and in many previous works). Do you think this problem is solvable?

**Limitations:**

The authors have adequately addressed the limitations and potential negative societal impact.

---

> ### Author Rebuttal · Authors · 2024-08-07
>
> Thank you very much for your thoughtful review. We are glad that the reviewer found our algorithm novel and the evaluation thorough and sound. We have tried to carefully address your questions. We hope the reviewer may consider raising their score in light of our response.
>
> ### Q1: Batch serving setting
>
> Thank you for this question. As you have correctly pointed out, Sequoia gives the largest gains for small batch sizes. This is because at small batch sizes, decoding is more memory-bound, and thus one can afford to verify a large tree without increasing verification time; by discovering the optimal structure for this large tree, Sequoia gives large speedups over baselines. Conversely, for larger batch sizes, the speculation budget is smaller, and the relative gains of Sequoia relative to other speculation tree structures (chains, k-ary trees, etc) is much smaller—we show this in Figure 1(a). In a real inference system with continuous batching, one could use Sequoia to determine the optimal tree for any batch size (as in Liu et al, 2024 [1]), thus attaining strong performance across the board. Furthermore, as you correctly pointed out, Sequoia can be combined with more powerful draft models (like Eagle) to attain even larger speedups (discussed briefly in Section 3.1.1).
>
> [1] Xiaoxuan Liu, Cade Daniel, Langxiang Hu, Woosuk Kwon, Zhuohan Li, Xiangxi Mo, Alvin Cheung, Zhijie Deng, Ion Stoica, Hao Zhang. Optimizing Speculative Decoding for Serving Large Language Models Using Goodput. CoRR abs/2406.14066, 2024.
>
> ### Q2: Performance decreasing with temperature
>
> Thank you for your insightful question!
>
> In this work, we use sampling without replacement to make the performance more robust across temperature. However, you are correct that decreasing is still observed when temperature goes up. Generally, high temperature means a higher degree of randomness, which is more difficult for draft models to guess. Improving the performance at higher temperatures is an interesting direction for future work—perhaps methods that better align draft and target models (e.g., via distillation [2]), or more advanced sampling algorithms [3, 4], could yield improvements here.
>
> [2] Zhou et al. DistillSpec: Improving Speculative Decoding via Knowledge Distillation. ICLR 2024.
>
> [3] Sun et al. Block Verification Accelerates Speculative Decoding. Efficient Systems for Foundation Models workshop, ICML 2024.
>
> [4] Qin et al. Multi-Token Joint Speculative Decoding for Accelerating Large Language Model Inference. Arxiv 2024.

---

> > ### Comment · Reviewer_yRtD · 2024-08-09
> >
> > I thank authors for their detailed response. I will maintain my score.

---

### Official Review · Reviewer_XEn9 · 2024-07-11

**Soundness:** 3
**Presentation:** 3
**Contribution:** 3
**Rating:** 7
**Confidence:** 2

**Summary:**

The paper proposes SEQUOIA, an algorithm designed to improve the efficiency of serving large language models (LLMs) through scalable and robust speculative decoding. The SEQUOIA algorithm introduces a dynamic programming method to construct optimal token trees for speculation, enhancing scalability. Additionally, a novel sampling and verification method ensures robustness across various decoding methods (top-k sampling, top-p sampling, temperature, ...). Empirical results demonstrate 4.04x speedup in a small model and 9.5x speedup in a large model with offloading inference.

**Strengths:**

- The simplest tree structure is a single chain (i.e., list) and the most complicated tree structure is k-ary full tree. SEQUOIA finds sweet spots between them, utilizing dynamic programming
- SEQUOIA maintains high hit-ratio across different sampling methods. This robustness make speculative decoding more practical.
- Paper presents extensive experimental results including ablation studies.

**Weaknesses:**

- Building optimal tree is more time-consuming than previous approaches (k-ary tree or k-independent sequences)
- Due to this, the batch size that can benefit from speculation might be much smaller compared to other methods

**Questions:**

- In Figure 4, why SpecInfer shows the opposite trend compared to Sequoia & Top-k sampling? In other words, Why it shows higher speedup when the temperature increases?
- It seems that SEQUOIA needs to conduct the dynamic programming algorithm every iteration. How long does that process take?

**Limitations:**

No additional limitations exist.

---

> ### Author Rebuttal · Authors · 2024-08-07
>
> Thank you very much for your thoughtful review. We are glad that the reviewer appreciates our dynamic programming based tree search algorithm as well as the robustness of sampling and verification algorithms. We have tried to carefully address your questions. We hope the reviewer can consider raising your score in light of our response.
>
> ### Q1: Time cost for dynamic programming and building optimal tree
>
> Thank you for raising this concern. The time for building a Sequoia tree has two parts:
>
> - *Offline Dynamic Programming*: Fixing the draft/target model and the workload, our dynamic programming can be conducted before inference (or offline). It is a one time preprocessing effort and does not introduce extra overhead during inference. Furthermore, our dynamic programming can run very fast, taking less than 20s to generate a tree as large as 1536, and less than 5s for trees of size 64 or 128.
>
> - *Tree building during inference*: Time for building the tree is decided by the inference cost of the draft model and the depth of the tree (as we analyzed in Section 4.1: Hardware Optimizer). In this sense, building a Sequoia-optimal tree of depth D should cost the same time as building a k-array tree of depth D.
>
> ### Q2: Sequoia’s performance at various batch sizes
>
> Thank you for this question. As you have correctly pointed out, Sequoia gives the largest gains for small batch sizes. This is because at small batch sizes, decoding is more memory-bound, and thus one can afford to verify a large tree without increasing verification time; by discovering the optimal structure for this large tree, Sequoia gives large speedups over baselines. Conversely, for larger batch sizes, the speculation budget is smaller, and the relative gains of Sequoia relative to other speculation tree structures (chains, k-ary trees, etc) is much smaller—we show this in Figure 1(a). In a real inference system with continuous batching, one could use Sequoia to determine the optimal tree for any batch size (as in Liu et al, 2024 [1]), thus attaining strong performance across the board.
>
>
> [1] Xiaoxuan Liu, Cade Daniel, Langxiang Hu, Woosuk Kwon, Zhuohan Li, Xiangxi Mo, Alvin Cheung, Zhijie Deng, Ion Stoica, Hao Zhang. Optimizing Speculative Decoding for Serving Large Language Models Using Goodput. CoRR abs/2406.14066, 2024.
>
>
> ### Q3: Different trends for SpecInfer, Top-K and Sequoia
>
> Thank you for raising this thoughtful question!
>
> In short, SpecInfer’s performance gets worse at low temperature, whereas Sequoia and Top-K sampling improve at lower temperatures, because these methods use different sampling and verification algorithms. SpecInfer uses sampling with replacement, Top-K uses top-k sampling and Sequoia uses sampling without replacement. As a result, at low temperatures SpecInfer is the only method that would repeatedly sample an incorrect token, leading to poor performance.
>
> We will now give a more detailed answer to your question, by first discussing how top-k sampling and SpecInfer sampling compare at both low and high temperatures, and then explaining how Sequoia is able to get “the best of both worlds” and perform well across both low and high temperatures.
>
> **In a low temp regime**, the token with largest probability in the target model’s output will take up over 95%. This token is very likely to appear in the top-k tokens of draft models (even if it is not the top-1 token). In this case, Top-K sampling can easily get accepted. However, for SpecInfer’s sampling with replacement, since the temperature is low, SpecInfer will keep drafting the same token. When the drafted token is not exactly the top-1 token of the target model, SpecInfer will get rejected. To summarize, in the low temp regime, Top-K requires the top-1 token of target model is among the top-k tokens of draft model, while SpecInfer requires the top-1 of the target model to be exactly the same as the top-1 of the draft model, leading to worse performance. Top-K method benefits from its cover property (Section 3.2.2).
>
> **In a high temp regime**, let’s consider an extreme case (temp $\rightarrow \infty$), i.e. the output of target model and draft model are totally random. In this case, it’s nearly impossible for Top-K to get accepted, as its total acceptance chance is just K / V, where V is the vocabulary size and K is the number of proposed tokens. However, for SpecInfer, all the draft tokens will be accepted, since the target and draft token probability is the same (1/V). SpecInfer method benefits from its optimal transport property (Section 3.2.2).
>
>
> Sequoia with sampling without replacement, behaves more like Top-K in low temp and more like SpecInfer in high temp as it has both cover property and optimal transport property. That’s the reason why Sequoia is able to perform well across a wide range of temperatures. The trend of acceptance rates of three methods is shown in Figure 3, which verifies our above claims.

---

### Official Review · Reviewer_qoEg · 2024-07-12

**Soundness:** 3
**Presentation:** 3
**Contribution:** 2
**Rating:** 6
**Confidence:** 3

**Summary:**

This paper proposed an improvement on the tree-based speculative decoding methods to make the accepted tokens scale roughly in logarithm to the number of tokens generated by the draft model.

The author provided theoretical and empirical justifications for the tree construction and verification procedures.  The experiments shows this method has a good speedup over naive method, and it can generate more tokens each time.

**Strengths:**

1. The paper has a strong theoretical guarantee for the scalability.

2. The method works for different scenarios with different temperature.

3. The dynamic programming problem can be pre-computed offline.

**Weaknesses:**

1. The differene with existing tree-based method in algorithm is marginal.
2. The speedups are compared against naive methods.  The speedup over SOTA is not provided.

**Questions:**

1. Could the author provide more reason why they don't compare with multiple draft model version of SpecInfer?
2. Could the author provide more justification for the validity of the positional acceptance assumption?  What's the impact of this assumption theoretically and empirically?  What is the distribution being used if the acceptance probability does not dependent on the token t?

**Limitations:**

The limitations are acknowledged but not fully addressed.

---

> ### Author Rebuttal · Authors · 2024-08-07
>
> Thank you very much for your thoughtful review. We are glad that the reviewer found our work to have strong theoretical guarantees for scalability and also have good empirical results. Also thanks for noticing that our dynamic programming algorithm can be pre-computed offline. We have tried to carefully address your questions. We hope the reviewer can consider raising your score in light of our response.
>
> ### Q1: Differences between Sequoia and existing tree-based methods
>
> Thank you for raising this question.
>
> Sequoia is designed to be a scalable and robust speculative decoding method that can yield large speedups in settings where the speculation budget is large (e.g., small batch serving, and offloading).  In these settings, we show how we can optimally construct very large trees (e.g., 128 tokens for on-chip, and 768 for offloading), and that the number of accepted tokens keeps growing as the tree size grows—in Figures 1,4,6 (and theorem 3.6) we show that the number of accepted tokens can grow roughly logarithmically with the number of speculated tokens in the tree. We believe that Sequoia could be even more powerful on future hardware—the gap between computation and memory bandwidth is getting higher (see Figure 1), thus allowing a larger speculation budget.
>
> Previous works like SpecInfer, Eagle, and Medusa apply shallow and small trees without studying the problem of scalability.  By leveraging Sequoia’s tree search algorithm, our empirical results show that Sequoia scales well to arbitrary tree sizes. On the offloading setting on L40, which has a large budget, Sequoia can outperform SpecInfer by 51% on average.
>
> In our work, in addition to improving the structure of the tree, we show that an improved sampling and verification algorithm can outperform others (SpecInfer, TopK) in terms of scalability and robustness. As described in Figure 4 and Table 5 in appendix, we can see that Sequoia achieves the largest speedups across all temperatures, attaining up to **1.65×** and **1.27×**  speedup relative to SpecInfer and top-k sampling, respectively.
>
>
> ### Q2: Speed up over SOTA
>
> Thank you for raising this issue. In the last columns of Table 1/2, we have presented the speedups of SpecInfer, which is a very strong baseline.
>
> In Table 1, we show that compared to SpecInfer, Sequoia attains speedups of 5% to 30% in the A100 on-chip setting (avg 22%). In Table 2, we show that Sequoia attains speedups over SpecInfer of  36% to 62% in the L40 offloading setting (avg 51%). More A100 on-device results can be found in Table 4 in appendix. We will add relative speedups compared to SpecInfer in the revised version.
>
> In addition, since our method does not improve/train draft models, we only compare with draft model agnostic baseline. And the draft model improvement methods, like Eagle, Glide, Medusa are orthogonal to us and can be combined to achieve a better performance.
>
>
> ### Q3: Reasons for not comparing with multiple-draft version of SpecInfer
>
> Thank you for raising this problem! We have the following three reasons.
> 1. *Draft model availability*: Often only a single draft model is available (e.g., Llama3-8B as a draft model for Llama3-70B), and it would thus require significant time/energy for practitioners to train additional draft models.
> 2. *Performance*: As compared in [SpecInfer](https://arxiv.org/pdf/2305.09781v3) (Appendix A, Table 4), most of the time one draft model outperforms multiple draft models.
> 3. *System*: SpecInfer serves each draft model on one GPU. However, every experiment in Sequoia is conducted on a single GPU, which means if we want to use the multiple-draft version, we need to run these draft models sequentially. This will further reduce the performance of the multi-draft version of SpecInfer.
>
>
> ### Q4: Positional Acceptance Assumption
>
> **Validity of the positional acceptance assumption**
>
> The positional acceptance assumption states that the probability of a verification algorithm
> accepting a token $t$ which is the $k^{th}$ child of an already accepted token depends only on the value of $k$. This is a simplifying assumption, given that the probability of a specific token getting accepted additionally depends on the token and context. However, we can consider the average acceptance rates, across many tokens/contexts, for sampled tokens, as a function of their position $k$.  In practice, we observe that this average is quite stable. For example (JF68M for Llama-2-7b, CNN, T=0.6, width=8), we show that when we measure the average acceptance rate vectors across 10 different groups of 200 prompts each, the variance across these groups is quite small..
>
> - Variance Vector: [0.0067, 0.0030, 0.0024, 0.0015, 0.0010, 0.0011, 0.0008, 0.0007]
> - Acceptance rate Vector: [0.5608, 0.1077, 0.0539, 0.0343, 0.0236, 0.0186, 0.0146, 0.0122]
> - Relative Error [1.2%, 2.8%, 4.4%, 4.5%, 4.2%, 5.9%, 5.3%, 5.4%]
>
>
> **Impact of positional acceptance assumption**
> - *Theoretically*:  we pointed out that, in tree based speculation methods, the probability of getting accepted is a function of the position of the speculated token. As a result, the optimal tree cannot be a balanced one (e.g. k-independent chains). This is also the intuition why we need and why we can search for an optimal tree.
> - *Empirically*: With expected acceptance probability for each position (i.e. our acceptance vector, vector P in algorithm 1), we can pre-compute expected accepted tokens to search for the optimal tree structure (algorithm 1).
>
>      For each experiment, we sample a subset of 200 sentences to calculate acceptance vectors and feed into algorithm 1 for tree searching.

---

### Official Review · Reviewer_uMCB · 2024-07-13

**Soundness:** 4
**Presentation:** 3
**Contribution:** 3
**Rating:** 7
**Confidence:** 4

**Summary:**

In this paper, the authors propose a novel speculative decoding algorithm to accelerate LLM generation. By leveraging the positional acceptance assumption and dynamic programming, they can determine an optimal tree topology with the tree size and depth. The experiments demonstrate that the proposed method outperforms previous works across various settings.

**Strengths:**

1. The paper is well-written and easy to follow.
2. The proposed method is supported by robust theoretical analysis and algorithm design.
3. Strong and comprehensive empirical results validate the proposed methods.

Overall, this paper demonstrates novelty, soundness, and a significant contribution to the field.

**Weaknesses:**

1. The experimental setup for offloading is not clearly explained. It is unclear if the draft model is placed on a GPU while part of the target model is on the GPU and the other part on the CPU. If this is the case, the speedup is understandable since the throughput of the target model would be very low, and the draft model can run very fast.
2. The comparison with the SpecInfer algorithm may not be fair. In Table 1, the size of the SpecInfer tree is 5x8=40, which is much smaller than the Sequoria tree. On the other hand, in Table 2, the size of the SpecInfer tree is 16x48=768, the same as the Sequoria tree. However, the length of SpecInfer would be very long (48) in this case, which seems impractical.

**Questions:**

1. What is the standalone throughput of the draft and target models under different configurations? This information would help readers understand where the speedup comes from.
2. How do you determine the best configuration of SpecInfer for comparison?
3. How accurate is the acceptance rate vector measuring with 200 examples?
4. Should the dynamic programming algorithm also consider the standalone throughput of the target and draft models? For instance, if the draft model is exactly the same as the target model, each element in the acceptance rate vector would be close to one, as the draft model can generate the same output as the target model. In this case, the tree will keep growing, but the more tokens the draft model generates, the slower it will become.

**Limitations:**

Please refer to the previous sections.

---

> ### Author Rebuttal · Authors · 2024-08-07
>
> Thank you very much for your thoughtful review. We are glad that the reviewer found our work **easy-to-follow** and having **comprehensive empirical results as well as theoretical analysis**. We have tried to carefully address your questions. We hope the reviewer can consider raising your score in light of our response.
>
> ### Q1: Clarifying the offloading setting
> Thank you for pointing out our unclear description. In our offloading setting, we performed a layer-wise offloading for the target model (default setting of deepspeed-zero) and the draft model is on-device.  So your understanding is correct, in this setting, the draft model is very fast (24.2ms) compared to the offloaded target model (5.7s).  With Sequoia, we can accelerate offloading based decoding from 5.7s/token to 0.6s/token, which is more tolerable for running a large model on a consumer GPU. Furthermore, the setting of offloading (a big gap between FLOPs and memory bandwidth) also simulates the trend of future hardware, as shown in Figure 1(a).
>
> ### Q2: Comparison with SpecInfer
> Thank you for your question about our SpecInfer baseline. Below we show, through 3 sets of experiments, that Sequoia outperforms SpecInfer when we perform thorough sweeps for the SpecInfer tree configuration:
>
> **(1) Sweep of SpecInfer tree structure**
>
> Here, we use JF68M as the draft model, and Llama2-7b as the target, and sweep a very wide range of SpecInfer tree structures (for both greedy and stochastic):
>
> Greedy Decoding (T=0.0):
>
> |Width/Depth| 1       | 2       | 4      | 8      | 16    | 32    | 64     | 128  |
> |---------|----|----|----|----|----|----|----|----|
> |1        | N/A| N/A| N/A|3.09x|3.14x|2.75x|1.94x| 1.19x|
> |2        |N/A | N/A|2.95x|3.36x|3.46x|2.69x|1.74x|    |
> |4   	    |N/A |2.40x|3.14x|3.46x|3.41x|2.47x|    |    |
> |8 	    |1.88x|2.44x|3.14x|**3.70x**|3.03x|    |    |    |
> |16 	    |2.00x|2.55x|3.27x|3.14x|    |    |    |    |
> |32       |1.86x|2.57x|2.81x|    |    |    |    |    |
> |64       |1.92x|2.22x|    |    |    |    |    |    |
> |128      |1.68x|    |    |    |    |    |    |    |
>
> Stochastic Decoding (T=0.6):
> |Width/Depth| 1       | 2       | 4      | 8      | 16    | 32    | 64     | 128  |
> |---------|----|----|----|----|----|----|----|----|
> |1        | N/A| N/A| N/A|2.08x|1.87x|1.48x|1.11x| 0.69x|
> |2        |N/A | N/A|2.14x|2.20x|1.89x|1.46x| 1.07x|    |
> |4   	    |N/A |1.99x|2.30x|2.28x|1.95x|1.53x  |    |    |
> |8 	    |1.73x|2.09x|2.42x|**2.42x**| 2.14x |    |    |    |
> |16 	    |1.78x|2.07x|2.41x|2.18x|    |    |    |    |
> |32       |1.78x|2.08x|2.24x|   |    |    |    |    |
> |64       |1.73x |2.04x  |    |   |    |    |    |    |
> |128      |1.61x | | ||    |    |    |    |
>
> Sequoia attains speedups of 4.04x (greedy) and 3.18x (stochastic), outperforming all tree configurations of SpecInfer, including 5x8 tree (3.45x greedy, 2.47x stochastic).
>
> **(2) More SpecInfer results for A100**
>
> For on-device settings, we add the results for the 8x8 tree and 16x8 tree (SpecInfer) as follows,
>
> Greedy Decoding (C4, T=0)
> |Draft|target|  Tree Config |   Sequoia  |  SpecInfer(5x8)   | SpecInfer(8x8)  |  SpecInfer(16x8) |
> |----|----|----|----|----|----|----|
> |JF68M| Llama-2-7b|     (128,10)    |**4.04x**|    3.45x|   3.70x| 3.16x|
> |JF68M  |  Llama-2-13b  |    (64,9)      |  **3.73x** | 3.30x |             3.10x| 2.4x|
> |SL1.3B|     Vicuna-33B  |      (64,6)     | **2.27x** |          1.83x| 1.73x|1.45x|
>
> Stochastic Decoding (C4, T=0.6)
>
> |Draft|target|  Tree Config|    Sequoia  | SpecInfer(5x8)   | SpecInfer(8x8)  |  SpecInfer(16x8)  |
> |----|----|----|----|----|----|----|
> |JF68M| Llama-2-7b| (128,7)  | **3.18x**|    2.47x|   2.45x|  2.18x|
> |JF68M  |  Llama-2-13b  |  (64,7)  |**3.19x**  | 2.48x |             2.42x|1.81x|
> |SL1.3B|     Vicuna-33B  |   (64,6)   |**2.16x**     |          1.64x| 1.52x| 1.32x |
>
> SpecInfer’s performance already degrades by enlarging the tree from 5x8 to 8x8. For SpecInfer, although the accepted tokens will marginally increase, the cost of verification/draft proposing will increase more.
>
> **(3) More SpecInfer results for L40 offloading**
>
> In Table 2 of the submission, we compare Sequoia trees of size 768 to SpecInfer trees of size 768, composed of 16 independent sequences of length 48 ("16x48").  Here we additionally compare to SpecInfer trees of shape 32x24 and 24x32:
>
> |Draft|target|  Tree Config| Sequoia | SpecInfer(16x48) |SpecInfer(32x24)|SpecInfer(24x32)|
> |----|----|----|----|----|----|----|
> |Llama-2-7b| Llama-2-70b|  (768, 18)|  **8.4x** (9.91) | 5.2x (7.03)|  5.5x (6.82) |  5.2x (6.66) |
>
> We will include a larger sweep in the revised version (these experiments are time consuming and require specific CPU/PCIEs that are often not available on cloud servers). Thank you for understanding!
>
> ### Q3: Standalone throughput of the draft and target models
> Thank you for pointing out this missing part. We will include the numbers in the revised version.
>
> JF68M:	0.5ms
>
> Llama-2-7b:	24.2ms
>
> Llama-2-70b:	5.7s
>
> Our system is implemented in Huggingface with CUDAGraph and Static KV Cache (not as optimized as frameworks such as vLLM and deepspeed).
>
> ### Q4: Accuracy of acceptance vector measurement
> Empirically, this measurement is accurate. Below, we show that when we measure the average acceptance rate vectors across 10 different groups of 200 prompts each, the variance across these groups is quite small:
>
> Setting: JF68M for Llama-2-7b, CNN, T=0.6, width=8
>
> - Variance Vector: [0.0067, 0.0030, 0.0024, 0.0015, 0.0010, 0.0011, 0.0008, 0.0007]
> - Acceptance rate Vector: [0.5608, 0.1077, 0.0539, 0.0343, 0.0236, 0.0186, 0.0146, 0.0122]
> - Relative Error [1.2%, 2.8%, 4.4%, 4.5%, 4.2%, 5.9%, 5.3%, 5.4%]
>
> ### Q5: dynamic programming algorithm should consider draft/target throughput
>
> Yes. We discussed this in Section 4.1 (Hardware Aware Optimizer). In the case you mentioned, the optimizer will choose the shallowest tree (depth = 0) even if the acceptance rate is high, which means we do not use speculative decoding at all.

---

> > ### Comment · Reviewer_uMCB · 2024-08-12
> >
> > Thank you for the responses. The authors have provided sufficient explanations and additional results, which have addressed most of my concerns.
> >
> > One additional limitation that comes to my mind is the context length: in this work, the context length is relatively short (128). It is unclear how it would perform with a longer context length (up to 8k). This could be explored in future work.

---

> > > ### Author Response · Authors · 2024-08-13
> > >
> > > Thank you for your response and insightful question!
> > >
> > > For long context serving, the memory bottleneck shifts from model parameters to KV cache. The increasing in context length is **orthogonal** to Sequoia since it almost does not increase the arithmetic intensity of decoding process, thus not reducing speculation budget. A recent work, Sun et al 2024 [1] discusses self-speculation for long contexts. We also conducted a simulation based on their methods, finding that Sequoia can help to accept about **30%** more tokens in their offloading setting for Llama-2-7B-128K (on PG19) than k-array trees with 256-512 speculation budget, demonstrating Sequoia's scalability in this scenario. It’s an interesting future work to evaluate Sequoia on a wide range of contexts with various draft models.
> > >
> > > **Simulation Results**
> > > |Budget| 256 | 384| 512|
> > > |----|----|----|----|
> > > |Sequoia|15.2|16.1|16.6|
> > > |16-array-tree|12.1|12.4|12.6|
> > >
> > > The number stands for #accepted tokens.
> > >
> > > Thank you once again for your feedback.
> > >
> > > [1] Sun, Hanshi, Zhuoming Chen, Xinyu Yang, Yuandong Tian, and Beidi Chen. Triforce: Lossless acceleration of long sequence generation with hierarchical speculative decoding

---

### Author Rebuttal · Authors · 2024-08-07

We thank all the reviewers [**R1** (uMCB), **R2** (qoEg), **R3** (XEn9), **R4** (yRTd)] for their thoughtful and highly supportive feedback! We were glad that the reviewers found the work **novel and meaningful** [R1,R3,R4], believed our theoretical analysis was **detailed, robust and strong** [R1, R2, R4], felt the experimental results were **sound and showed good speed ups** [R1, R2, R3, R4], and found the presentation **easy to follow** [R1].  We have updated the paper to incorporate constructive suggestions, which will show in the revision. We summarize the major changes:

1. **Comparison with SpecInfer/Baselines** [R1, R2]:
    - We added relative speed up numbers over SpecInfer, achieving avg 22% for A100 and 51% for offloading.
    - We swept a wide range of tree configurations for SpecInfer for a more fair comparison. We added this part as an ablation.

2. **Positional Acceptance Assumption** [R1, R2]: To further clarify this assumption and its correlation with our algorithm, we added a discussion about the acceptance rate vector we measured for each experiment and its variance (1~5%, indicating the measurement is accurate).

---

### Decision · Program_Chairs · 2024-09-25

**Decision:**

Accept (spotlight)

**Comment:**

The reviewers unanimously praise the paper's strong theoretical guarantees, novelty and extensive evaluation.
Furthermore, the reviewers either acknowledged the rebuttal addresses their concerns (uMCB) or that it is detailed and did not follow with more comments (yRtD). Given that and the initial support for the paper, there does not seem to be any outstanding issues regarding the paper. I have read it quickly and it appears to be of high quality and, importantly, the authors include discussion about limitations in the appendix. Considering all the things, and especially its theoretical grounding and resulting strong performance, it is a clear candidate for acceptance.

The only ask for the authors would be to incorporate as much as possible from the review process into a revision of their paper, since I expect a lot of those things would be interesting for potential readers as well.